# Quantifying the noise sensitivity of the Wasserstein metric for images

## Abstract

Wasserstein metrics are increasingly being used as similarity scores for images treated as discrete measures on a grid, yet their behavior under noise remains poorly understood. In this work, we consider the sensitivity of the signed Wasserstein distance with respect to pixel-wise additive noise and derive non-asymptotic upper bounds. Among other results, we prove that the error in the signed 2-Wasserstein distance scales with the square root of the noise standard deviation, whereas the Euclidean norm scales linearly. We present experiments that support our theoretical findings and point to a peculiar phenomenon where increasing the level of noise can decrease the Wasserstein distance. A case study on cryo-electron microscopy images demonstrates that the Wasserstein metric can preserve the geometric structure even when the Euclidean metric fails to do so.

## 1 Introduction

Optimal Transport (OT) provides a principled way to measure the distance between probability measures, capturing not only pointwise differences but also the underlying geometry of the data. Recent advances in computational approximation methods (Cuturi, 2013; Schmitzer, 2019) contributed greatly to the rising popularity of optimal transport across many domains, such as computer vision (Feydy et al., 2021), domain adaptation (Courty et al., 2017), and others. In imaging applications, the Wasserstein metric can be used to measure similarity by treating images as discrete measures on a grid, and assigning a point mass to every pixel, proportional to its value. One field where this approach is gaining popularity is in single-particle cryo-electron microscopy (cryo-EM), a domain characterized by extremely high noise levels, where OT-based methods have been successfully applied to fundamental tasks, including the alignment of 3D density maps (Riahi et al., 2022; Singer & Yang, 2024), the clustering 2D tomographic projections (Rao et al., 2020), and the rotational alignment of tomographic projections with heterogeneity (Shi et al., 2025). We believe that a major driver for this adoption is that, empirically, the Wasserstein metric appears more robust to noise than the standard Euclidean norm.

**Related work.** In generative modeling, OT-based metrics have inspired methods such as Wasserstein GAN (Arjovsky et al., 2017) and Wasserstein autoencoders (Tolstikhin et al., 2017) and flow matching (Lipman et al., 2022; Albergo & Vanden-Eijnden, 2023; Liu et al., 2022). The latter in particular has strong connections to OT in its dynamic formulation. Building upon this, recent variants of flow matching incorporate OT solvers into the training process (Tong et al., 2023; Chemseddine et al., 2025; Zhang et al., 2025; Mousavi-Hosseini et al., 2025). While our work does not target these models specifically, we believe that a better understanding of the noise robustness of optimal transport procedures is needed to understand why modern generative models work so well.

Many variants of OT such as partial optimal transport (Chapel et al., 2020; Raghvendra et al., 2024) and unbalanced optimal transport (Benamou et al., 2015; Chizat et al., 2018) have been proposed to address mass imbalance. While not strictly comparable to the our work, these works cover a different notion of robustness.

**Our contribution.** On the theoretical side, we provide quantitative bounds relating the signed Wasserstein cost (see equation 3) between noise-corrupted images and the signed Wasserstein cost between the clean images. Focusing on a Gaussian noise model with fixed mass and pixel-wise

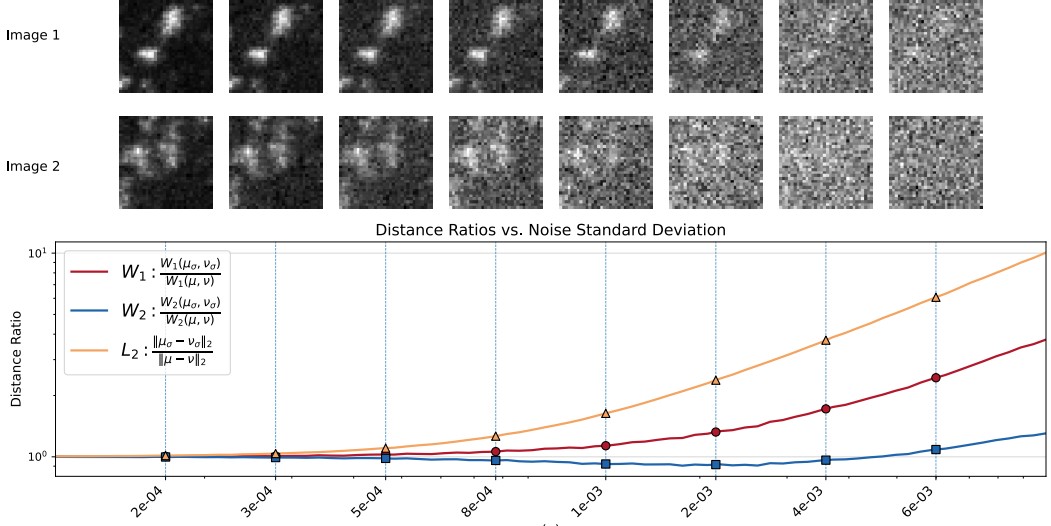

Figure 1: Distance ratios of $L^2, W_1$ and $W_2$ on a pair of noised images as a function of the noise level. $L^2$ diverges first, followed by $W_1$ and lastly, $W_2$ departs from the original distance between the images, exhibiting more noise robustness. Above each marker we show the pair of images that were compared using all 3 metrics. See Section 4.2 for more details.

standard deviation proportional to $\sigma$, we show that the signed $p$-Wasserstein metric between a noise-corrupted $n \times n$ picture and its clean counterpart gives rise to an error term that scales like $(n\sigma)^{1/p}$, see Theorem 3. For the 1-Wasserstein distance, considering a similar noise model, Theorem 4 establishes that the distance between two noisy pictures deviate at most by an order $\sigma n \log_2 n$ from the distance between the clean ones. Theorem 5 gives a bound for the case of two different measures and $p \geq 1$. We complement our theoretical results with simulations in Section 4, showcasing the properties of the signed Wasserstein distance in a variety of cases.

## 2 WASSERSTEIN OVER NOISED AND SIGNED MEASURES

**Wasserstein metric.** Consider two probability measures $\mu, \nu \in \mathcal{P}(\mathcal{X})$. For any $p \geq 1$ and given a ground cost $\mathsf{d} : \mathcal{X} \times \mathcal{X} \to \mathbb{R}_+$, the Wasserstein metric between $\mu$ and $\nu$ is defined as

$$W_p(\mu, \nu) := \left( \inf_{\pi \in \Gamma(\mu, \nu)} \int_{\mathcal{X} \times \mathcal{X}} \mathsf{d}(x, y)^p \mathrm{d}\pi(x, y) \right)^{1/p}, \tag{1}$$

where $\Gamma(\mu, \nu)$ is the set of measures with respective marginals $\mu$ and $\nu$.

Under mild conditions, see Santambrogio (2015, Theorem 1.39), the Wasserstein distance admits a dual formulation, i.e.,

$$W_p^p(\mu, \nu) = \sup_{\phi \in L^1(\mu)} \int_{\mathcal{X}} \phi(x) \mathrm{d}\mu(x) + \int_{\mathcal{X}} f^{\mathsf{d}}(y) \mathrm{d}\nu(y),$$

where $f^{\mathsf{d}}(y) := \inf_{x \in \mathcal{X}} \left( \mathsf{d}(x, y)^p - f(x) \right)$.

In the case of the 1-Wasserstein distance, the dual formulation further admits the simplified form

$$W_1(\mu, \nu) = \sup_{f \in \mathrm{Lip}_1(\mathcal{X})} \langle f, \mu - \nu \rangle, \tag{2}$$

where $\mathrm{Lip}_1(\mathcal{X})$ is the set of 1-Lipschitz functions with respect to $\mathsf{d}$ on $\mathcal{X}$. The dual formulations are particularly useful to study stability of optimal transport with respect to perturbations of the marginals $\mu$ and $\nu$.

**Extension to signed measures.** Some image modalities (such as cryo-EM) naturally involve negative pixels, but even in modalities where all pixels are positive, once pixel-wise noise is introduced, negative pixels may appear. Since we identify pixel values with point masses, to study the effect of pixel-wise noise we first need to explain how the Wasserstein metrics can be extended to support negative masses. The problem of generalizing OT to signed measures is not new; Mainini (2012) proposed to compute a Wasserstein-like distance between $\mu$ and $\nu$ by first constructing

$$S_{\mu,\nu} = \mu_+ + \nu_- \qquad \text{and} \qquad T_{\mu,\nu} = \nu_+ + \mu_-, \tag{3}$$

where we assume that the total mass of $\mu$ and $\nu$ are equal and $\mu_+$ ($\mu_-$, resp.) denotes the positive (resp. negative) part of $\mu$. Mainini (2012) then introduced the *signed Wasserstein cost*,

$$W_p^\pm(\mu,\nu) := W_p(\mu_+ + \nu_-, \nu_+ + \mu_-) = W_p(S_{\mu,\nu}, T_{\mu,\nu}), \tag{4}$$

which the author denoted by $\mathbb{W}_p(\mu,\nu)$. We note that $W_p^\pm$ is a metric for $p = 1$ but not for $p > 1$, since it does not satisfy the triangle inequality (Mainini, 2012, Proposition 3.4). The fact that the 1-Wasserstein metric combines nicely with the positive and negative parts can be deduced from equation equation 2. The absence of triangle inequality might further lead to surprising behaviors as seen in Figure 4.

Other approaches for generalizing the Wasserstein distance to signed measures were explored by Engquist et al. (2016). Further, the usefulness of signed Wasserstein costs as defined above starts to be acknowledged in the statistics literature as the recent preprint by Groppe et al. (2025) suggests.

**The issue of noise.** We model images as real-valued signals on a square grid $G_n$ of $n^2$ pixels which we identify with signed discrete measures. The aim of this work is to investigate how $W_p^\pm$ behaves when the images/measures $\mu$ and $\nu$ are corrupted. In particular, consider observing

$$\mu_\varepsilon := \mu + \varepsilon_\mu \quad \text{and} \quad \nu_\varepsilon := \nu + \varepsilon_\nu \tag{5}$$

and constructing

$$S_{\mu_\varepsilon,\nu_\varepsilon} := (\mu_\varepsilon)_+ + (\nu_\varepsilon)_- \quad \text{as well as} \quad T_{\mu_\varepsilon,\nu_\varepsilon} := (\nu_\varepsilon)_+ + (\mu_\varepsilon)_-. \tag{6}$$

Further set

$$C_S := \sum_{x \in G_n} (\mu_\varepsilon)_+(x) + (\nu_\varepsilon)_-(x) \quad \text{and} \quad C_T := \sum_{x \in G_n} (\nu_\varepsilon)_+(x) + (\mu_\varepsilon)_-(x). \tag{7}$$

Standardizing $S_{\mu_\varepsilon,\nu_\varepsilon}$ by $C_S$ and $T_{\mu_\varepsilon,\nu_\varepsilon}$ by $C_T$ is necessary to ensure that both measures have the same (unit) mass in the case where $\sum_{x \in G_n} \mu_\varepsilon(x) \neq \sum_{x \in G_n} \nu_\varepsilon(x)$. In the sequel, we will use the notation

$$\bar{S}_{\mu_\varepsilon,\nu_\varepsilon} := \frac{S_{\mu_\varepsilon,\nu_\varepsilon}}{C_S} \quad \text{and} \quad \bar{T}_{\mu_\varepsilon,\nu_\varepsilon} := \frac{T_{\mu_\varepsilon,\nu_\varepsilon}}{C_T}. \tag{8}$$

We aim at understanding the relationship between $W_p^\pm(\mu,\nu)$ and $W_p(\bar{S}_{\mu_\varepsilon,\nu_\varepsilon}, \bar{T}_{\mu_\varepsilon,\nu_\varepsilon})$. To put our analysis into context, consider the standard squared $L^2$ distance, a common metric for image comparison. In the presence of additive Gaussian noise with variance $\sigma^2$, the expected squared $L^2$ distance between a signal and its noisy version has a simple, direct relationship: it is exactly $n\sigma$. This metric, however, is local and insensitive to the underlying geometric structure of the signal. In contrast, the (signed) Wasserstein cost is claimed to capture this geometry, but its behavior under noise is far more complex to characterize. This paper aims to bridge that gap by providing a theoretical and empirical analysis of its robustness.

**Dyadic bound on the Wasserstein distance.** To get sharp estimates on the Wasserstein distance, the following proposition is particularly useful. This is Proposition 1 of Weed & Bach (2019), but on a domain with an arbitrary diameter (their formulation assumed $\text{diam}(S) = 1$). The bound is based on the construction of a coupling at various scales, managing the mass imbalance in subdomains. This construction yields sharp rates in a variety of cases.

**Proposition 1.** *Let $\{Q^k\}_{1 \leq k \leq k^*}$ be a dyadic partition of a set $S$ with parameter $\delta < 1$. Then, for probability measures $\mu$ and $\nu$ supported on $S$,*

$$W_p^p(\mu,\nu) \leq \text{diam}(S)^p \left( \delta^{pk^*} + \sum_{k=1}^{k^*} \delta^{p(k-1)} \sum_{Q_i^k \in \mathcal{Q}^k} |\mu(Q_i^k) - \nu(Q_i^k)| \right). \tag{9}$$

Recall that a dyadic partition of a set $S$ with parameter $\delta < 1$ is a sequence $\{\mathcal{Q}^k\}_{1 \le k \le k^*}$ possessing the following properties. First, the sets in $\mathcal{Q}^k$ form a partition of $S$. Further, if $Q \in \mathcal{Q}^k$, then $\mathrm{diam}(Q) \le \delta^k$. Finally, if $Q^{k+1} \in \mathcal{Q}^{k+1}$ and $Q^k \in \mathcal{Q}^k$, then either $Q^{k+1} \subset Q^k$ or $Q^{k+1} \cap Q^k = \emptyset$.

# 3 THEORETICAL CONTRIBUTIONS

Our main theoretical results are upper bounds in expectation on the effect that the noise has on the signed Wasserstein cost between images. To avoid boundary effects and simplify some of our analyses, we consider the pixel grid to have cyclic boundary conditions, i.e., the left–right and top–bottom edges wrap. With this choice, each pixel has the same number of neighbors.

While the Wasserstein metric naturally extends to non-probability measures, it still requires that both measures have the same mass, as described in the previous subsection. A standard i.i.d. noise model comes with the need of rescaling the pictures, which we study in the following section.

*Note that all proofs of the following results are collected in Appendix A.*

## 3.1 THE IMPACT OF RESCALING.

An important fact is that the signed Wasserstein distance, by construction, has an intricate non-linear behavior in terms of the noise when the mass of the latter is not fixed. By duality, observe that

$$W_p^p(\bar{S}_{\mu_\varepsilon, \nu_\varepsilon}, \bar{T}_{\mu_\varepsilon, \nu_\varepsilon}) = \sup_f \langle f, \bar{S}_{\mu_\varepsilon, \nu_\varepsilon} \rangle + \langle f^{\mathsf{d}}, \bar{T}_{\mu_\varepsilon, \nu_\varepsilon} \rangle \tag{10}$$

$$= \sup_f \frac{\langle f, S_{\mu_\varepsilon, \nu_\varepsilon} \rangle + \langle f^{\mathsf{d}}, T_{\mu_\varepsilon, \nu_\varepsilon} \rangle}{\sum_{x \in G_n} S_{\mu_\varepsilon, \nu_\varepsilon}(x)} + \left( \frac{1}{\sum_{x \in G_n} T_{\mu_\varepsilon, \nu_\varepsilon}(x)} - \frac{1}{\sum_{x \in G_n} S_{\mu_\varepsilon, \nu_\varepsilon}(x)} \right) \langle f^{\mathsf{d}}, T_{\mu_\varepsilon, \nu_\varepsilon} \rangle. \tag{11}$$

This decomposition shows that the optimal dual function must balance two objectives at the same time: the first one is the transport problem, and the second can be interpreted as a mass imbalance penalization. In the case of i.i.d. Gaussian noise, the result above can be refined to yield,

**Theorem 1.** *Consider two $n \times n$ images $\mu$ and $\nu$ having at least $\lambda n^2$, $\lambda \in (0, 1]$ nonzero pixels. Assume that $\varepsilon_\mu, \varepsilon_\nu$ are $\mathcal{N}(0_{n^2}, \sigma^2 I_{n^2})$. Recall the definition of $\bar{S}_{\mu_\varepsilon, \nu_\varepsilon}, \bar{T}_{\mu_\varepsilon, \nu_\varepsilon}$ in equation 8. Then,*

$$W_1^\pm(\bar{S}_{\mu_\varepsilon, \nu_\varepsilon}, \bar{T}_{\mu_\varepsilon, \nu_\varepsilon}) = \frac{1}{\sum_{x \in G_n} S_{\mu_\varepsilon, \nu_\varepsilon}(x)} \sup_{f \in \mathrm{Lip}_1} \left\langle f, S_{\mu_\varepsilon, \nu_\varepsilon} - T_{\mu_\varepsilon, \nu_\varepsilon} \left( 1 + \mathrm{O}_p\left( \frac{\sigma}{n} \right) \right) \right\rangle. \tag{12}$$

Even though the above result does not seem symmetric, we establish in the proof that

$$\sum_{x \in G_n} S_{\mu_\varepsilon, \nu_\varepsilon}(x) - T_{\mu_\varepsilon, \nu_\varepsilon}(x) = \mathrm{O}_p(\sigma n), \tag{13}$$

from which we deduce that the apparent absence of symmetry is merely an artifact of the proof.

In general, one can hope that the ratio $\sigma/n$ is small, so that the result suggests that understanding the quantity $\sup_f \langle f, S_{\mu_\varepsilon, \nu_\varepsilon} \rangle + \langle f^{\mathsf{d}}, T_{\mu_\varepsilon, \nu_\varepsilon} \rangle$ under a suitable choice of noise is a first step to take towards completely characterizing the impact of the noise.

## 3.2 NOISE MODEL

The previous section invites us to consider a noise model for which it is not necessary to rescale the measures. To this end, we will consider slightly correlated Gaussian noise where we identify each coordinate of the Gaussian noise vector with a point on the regular grid $G_n$.

**Assumption 1.** *Consider an image modeled as an $n \times n$ grid of pixels and set $m = n^2$. Assume that the noise vector $N = (N_1, \ldots, N_m)$ is drawn from a multivariate normal distribution $\mathcal{N}(0, \Sigma)$, where the covariance matrix $\Sigma$ is an $m \times m$ matrix defined as*

$$\Sigma_{ij} = \begin{cases} \sigma^2 & \text{if } i = j \\ -\frac{\sigma^2}{m-1} & \text{if } i \ne j. \end{cases} \tag{14}$$

Note that this noise model is equivalent to drawing the pixels independently from $\mathcal{N}\left(0, \sigma^2 m/(m-1)\right)$, calculating their mean, and then subtracting the mean from every pixel.

**Proposition 2** (Noise model properties). *In the context of Assumption 1, the following holds.*

1. *The marginal distribution for each component is $N_i \sim \mathcal{N}(0, \sigma^2)$.*

2. *The sum of the components is zero : $\sum_{i=1}^{M} N_i = 0$.*

This last property allows us to focus on the impact of the noise, while setting aside the questions pertaining to rescaling the measures whose behavior was captured in Theorem 1.

### 3.3 MULTISCALE $W_p$ BOUND ON A SINGLE IMAGE AND ITS NOISY VERSION

We shall begin by proving bounds in the particular case where we compare one image with a noise corrupted version of itself. We start with the case of $p = 1$.

**Theorem 2.** *Let $\mu : G_n \to [0,1]$ be a probability measure on the $n \times n$ unit grid $G_n$ with cyclic boundary conditions. Let $\varepsilon_1, \varepsilon_2$ be independent random signed measures on the grid that satisfy Assumption 1. Then*

$$\frac{n\sigma}{\sqrt{\pi}} \leq \mathbb{E}W_1^{\pm}(\mu + \varepsilon_1, \mu + \varepsilon_2) \leq \frac{2\sqrt{2}n\log_2 n}{\sqrt{\pi}}\sigma + \frac{n}{\sqrt{2\pi}}\sigma. \tag{15}$$

It is further possible to prove a result for $p > 1$. The rates differ substantially, as is clear from the following theorem.

**Theorem 3.** *Let $\mu : G_n \to [0,1]$ be a probability measure on the $n \times n$ unit grid $G_n$. Let $\varepsilon_1, \varepsilon_2$ be independent random signed measures on the grid that satisfy Assumption 1. For convenience, we again assume that $n = 2^\eta$, for $\eta \in \mathbb{N}$. Then, for $p > 1$ with $p \in \mathbb{N}$,*

$$\mathbb{E}\left[\left(W_p^{\pm}(\mu + \varepsilon_1, \mu + \varepsilon_2)\right)^p\right] \leq \frac{4\sqrt{2}}{\sqrt{\pi}}n\sigma. \tag{16}$$

*Therefore, by Jensen's inequality,*

$$\mathbb{E}\left[W_p^{\pm}(\mu + \varepsilon_1, \mu + \varepsilon_2)\right] \leq \left(\frac{4\sqrt{2}}{\sqrt{\pi}}n\sigma\right)^{1/p}.$$

*Remark* 1. In both theorems above, the upper bound can be improved by removing the factor $\sqrt{2}$ if only one image is corrupted by noise.

### 3.4 MULTISCALE $W_p$ BOUND ON TWO IMAGES AND THEIR NOISY COUNTERPARTS

We now consider the practically relevant setting where two different images are each corrupted by independent noise. Throughout, we assume that the noise model follows Assumption 1 and assume that both images have unit mass. Our object of interest is thus

$$W_p^{\pm}(\mu + \varepsilon_\mu, \nu + \varepsilon_\nu). \tag{17}$$

In the case $p = 1$, one obtains the following result.

**Theorem 4.** *Let $\mu, \nu : G_n \to [0,1]$ be two probability measures on the $n \times n$ unit grid $G_n$ with cyclic boundary conditions and let $\varepsilon_\mu, \varepsilon_\nu : G_n \to \mathbb{R}$ be signed noise measures that satisfy Assumption 1. For convenience we assume that $n = 2^\eta$, for $\eta \in \mathbb{N}$. Then*

$$\mathbb{E}\left[W_1^{\pm}(\mu + \varepsilon_\mu, \nu + \varepsilon_\nu) - W_1^{\pm}(\mu, \nu)\right] \leq \frac{4n\log_2 n + n}{\sqrt{\pi}}\sigma + \frac{\sqrt{2}}{n}. \tag{18}$$

The proofs of the three theorems above come from a multiscale upperbound on the Wasserstein distance that depends solely on the mass differences (here, the pixels intensities) and therefore enables to control the impact of the noise.

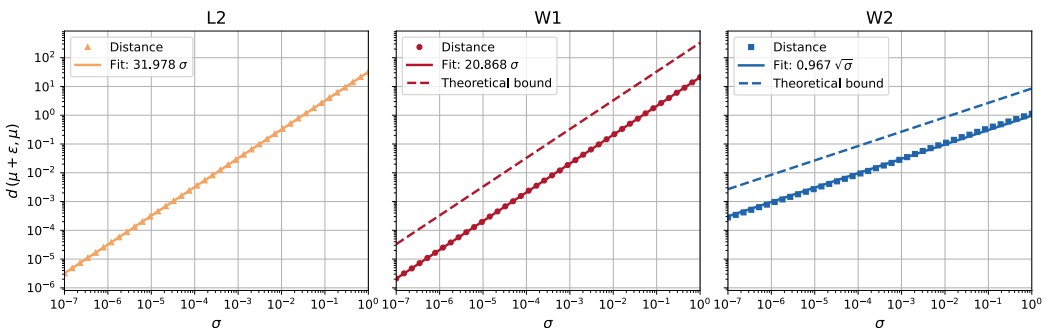

Figure 2: $W_1, W_2$ and $L_2$ (markers) distances plotted against their fits and their theoretical bounds from Theorems 2 and 3 (dashed lines).

Even though the Wasserstein 2-distance is often used in applications and has nice theoretical properties in the continuous setting —such as the Brenier–McCann theorem (Brenier, 1991), its signed counterpart does not enjoy the same metric properties as the signed Wasserstein 1-distance, as was already hinted at in the introduction.

This absence of triangle inequality underlies the particular form of the following result.

**Theorem 5.** *Let $\mu, \nu : G_n \to [0, 1]$ be two probability measures on the $n \times n$ unit grid $G_n$ with cyclic boundary conditions and let $\varepsilon_\mu, \varepsilon_\nu : G_n \to \mathbb{R}$ be signed noise measures that satisfy Assumption 1. For convenience we assume that $n = 2^\eta$, for $\eta \in \mathbb{N}$. Then,*

$$\mathbb{E}\big[W_p^\pm(\mu + \varepsilon_\mu, \nu + \varepsilon_\nu)\big] \leq \left(\frac{\sqrt{2}}{2}\right)^{1-\frac{1}{p}} W_1(\mu, \nu)^{\frac{1}{p}} + \frac{\sqrt{2}}{2}\left(\frac{4}{\sqrt{\pi}}n\log_2 n + \frac{2}{\sqrt{\pi}}n\right)^{\frac{1}{p}}\sigma^{\frac{1}{p}}. \quad (19)$$

The proof of this theorem requires to first relate the signed Wasserstein $p$ distance to a 1-Wasserstein distance between the uncorrupted measures by exploiting suboptimal couplings. This then enables to use a multiscale bound for the remaining part that mostly pertains to noise.

## 4 NUMERICAL EXPERIMENTS AND RESULTS

### 4.1 QUANTITATIVE VALIDATION OF NOISE SCALING

The first experiment we conduct aims to quantitatively measure how the distance between an image and its noisy counterpart scales when increasing noise variance. This allows for a direct comparison between the empirical behavior of each metric and the theoretical scaling laws derived in Theorem 3. The results are reported as Figure 2. All transport costs calculated are exact and were calculated by using the POT python package Flamary et al. (2021)

To this end, we performed 100 independent trials, each time selecting a new, random 32x32 pixel image from the DOTMark 1.0 MicroscopyImages dataset (Schrieber et al., 2017). For each image $\mu$, we generated a noisy version $\mu + \varepsilon$ by adding zero-sum noise $\varepsilon$ satisfying Assumption 1, with variances ranging from $10^{-7}$ to 1. We computed the difference between the original image and the noisy one for $L^2$, $W_2$ and $W_1$ identifying the image with the torus. This empirical result, where the $W_2$ distance scales with an exponent of approximately 0.5, suggests that the bound derived in Theorem 3 correctly captures the behavior of the signed 2-Wasserstein as a function of the noise variability $\sigma$.

### 4.2 VISUALIZING ROBUSTNESS OF INTER-IMAGE DISTANCES

We now investigate how well the different metrics preserve the original distance between two images when the latter are progressively corrupted by noise. For this experiment, we selected two distinct $32 \times 32$ pixel images from the DOTMark dataset and simultaneously corrupted them with different

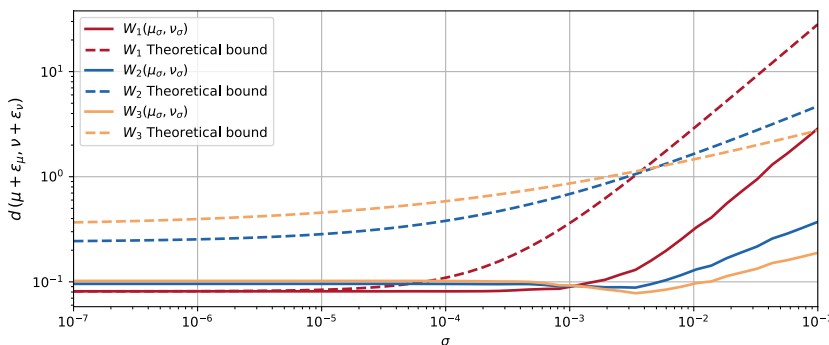

Figure 3: Distances between two randomly sampled images from the DOTMark microscopy dataset, both being noised with noise sampled from the zero-sum normal distribution, in dashed (matching colors) we have the bounds for each $p$ from Theorem 3.

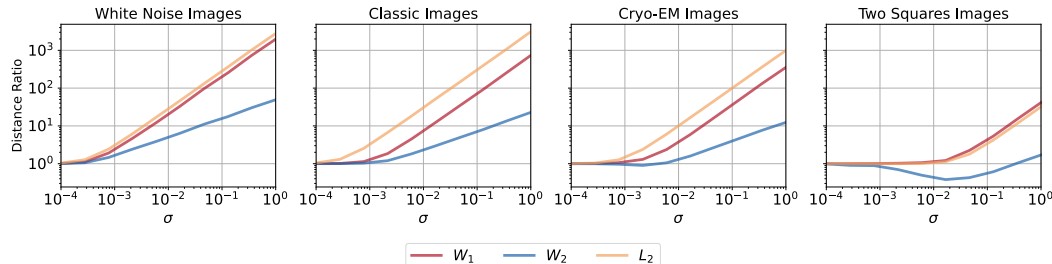

Figure 4: Ratios of the distance between the noisy images and the original images. Left: White noise images. Center left: Classic images. Center right: Images from the field of cryo-em. Right: Images of two squares which have all the mass centered in one place.

instances of zero-sum additive noise with a standard deviation ranging from $10^{-7}$ to $10^{-1}$. At each noise level, we computed the $W_1, W_2$ and $L^2$ distances between the two noisy images. The results were averaged across 100 experiments. To evaluate stability, we computed a "Distance Ratio" by dividing the distance between the noisy images by the constant distance between the original, clean images. A ratio that remains close to 1 indicates that the metric's measurement reflects the underlying signals rather than the noise. The output is displayed on Figure 1, which we already exhibited in the introduction. On that figure, the top panel visually depicts the degradation of the images as noise increases, while the main plot shows the distance ratio for each metric. The $L^2$ ratio (salmon-colored line) is the first to sharply diverge from 1, showing that the measured distance is quickly dominated by the noise. The $W_2$ ratio (blue line) is the most stable, remaining closest to the ideal ratio of 1 for the largest range of noise levels. This experiment serves as a practical illustration of the scaling laws: as the $W_2$ distance grows more slowly with noise, the underlying distance between the clean signals is better preserved.

**Visualisation of the bound of the inter-image distance.** To assess the bound established in Theorem 5, we have plotted the distance between two cryo-EM images being gradually corrupted by noise with the same parameter $\sigma$. We see in Figure 3 how tight the bound might be for $W_1$ (in the case of small noise) while it seems to not be tight for $W_2$ and $W_3$. We postulate that this is because the images used in this experiment are far from the "worst case scenario" in which where the images are very similar to each other, or very far apart. The characterization of these scenarios where the bound might be tighter comes from the analysis reported on Figure 7.

**Characterizing metric behavior across image types.** While $W_2$ is robust, its behavior is not uniform. The purpose of the next experiment is to explore how the metrics' robustness varies across different classes of images and to highlight a key nuance of the signed $W_2$ metric.

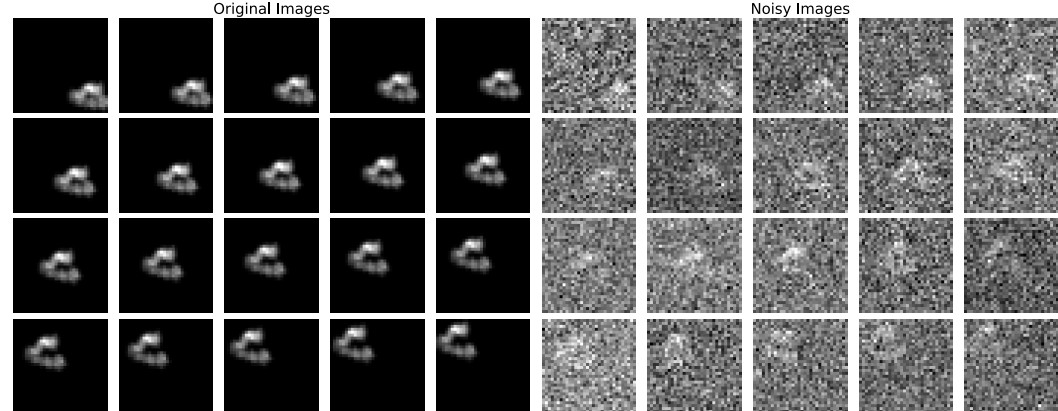

Figure 5: Projection images of the E. Coli hsp90 molecule, with and without noise.

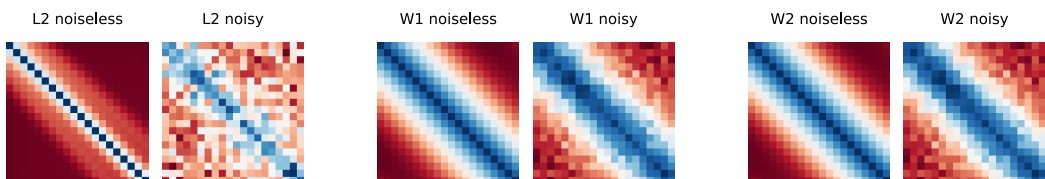

Figure 6: Top panel: Distance matrices between the different structures in the absence of noise. Bottom panel: averaged distances over 100 experiments. A more gradual blue-to-red gradient is better, $W_2$ performs best.

We repeated the distance ratio experiment from the previous section on four distinct image classes: white noise, typical cryo-EM projections, classic microscopy images, and synthetic images of two widely separated squares. These classes represent types of images which are ranging from pure noise, The results are shown in Figure 4. In this figure, $W_2$ is shown to scale favorably. However, it also exhibits an peculiar lack of monotonicity w.r.t. the noise level. A phenomenon that we explore in Section 4.4.

## 4.3 ANALYSIS OF CRYO ELECTRON-MICROSCOPY IMAGES

Single-particle cryo-electron microscopy (cryo-EM) is a method for reconstructing the 3D structure of proteins and other large molecules. In this method, samples of a molecule of interest are frozen and then imaged using a transmission electron microscopy. This results in many thousands of tomographic projections of the target molecule. The positions and orientations of the individual molecules is typically unknown and the images have extremely high level of noise. Nonetheless, sophisticated computational methods were successful in recovering many different high-resolution 3D structures. Many important challenges remain. In particular, the reconstruction of flexible macromolecules with continuous degrees of freedom. See Bendory et al. (2020) for a survey of the computational challenges in cryo-EM. In this section, we wish to demonstrate the potential benefit of Wasserstein metrics in the high-noise cryo-EM regime to the difficult task of recovering continuous conformational manifolds (Kileel et al., 2021). To this end, we generated 20 different projections of the E. Coli hsp90 protein (Shiau et al., 2006) in different conformational states using cryoJAX (O'Brien et al., 2025). The location of the protein was shifted and the pictures were corrupted with high levels of noise to mimic the poor signal-to-noise ratio in real cryo electron microscopy images. For simplicity, all the images were normalized to sum to one. The goal is to assess how well each metric can recover known geometric relationships between particle images that undergo rotation and translation. To illustrate the difficulty of the task, Figure 5 shows a sample of the original, clean images alongside their noisy counterparts. For each metric, we compute all the pairwise distances, resulting in a 20x20 matrix which we can see in Figure 6. The top row shows the ground-truth distance matrices from the clean images, reflecting the structured of the transformations. The bottom row

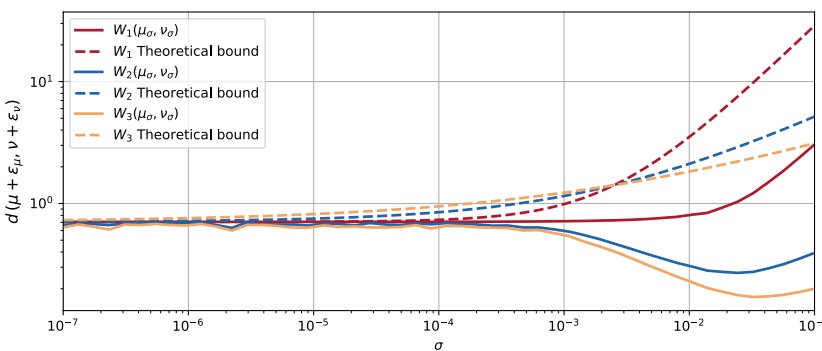

Figure 7: Distances between two images, one has a mass of 1 in (8,8) and the other in (24,24) and zero everywhere else, both being noised with noise sampled from the zero-sum normal distribution, in dashed (matching colors) we have the bounds for each $p$ from Theorem 5.

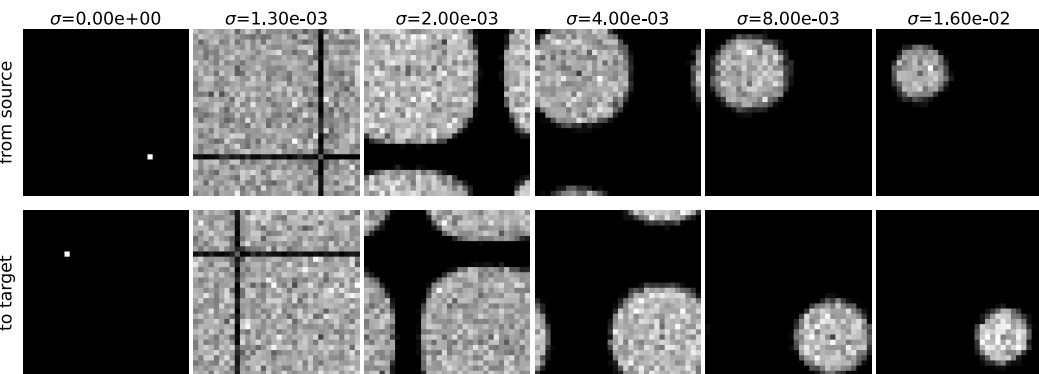

Figure 8: Top - Source image, and where the mass of the original pixel (8,8) goes. Bottom - Target image, and where the mass of the target pixel (24, 24) comes from for the optimal transport map between noised versions of the single pixel images.

shows the matrices computed from their noisy counterparts. Under heavy noise, the $L^2$ distance matrix degrades into a random pattern, losing the original geometric structure. In contrast, the $W_2$ distance matrix preserves the global diagonal structure of the ground-truth matrix. This result is also described by our theoretical results.

### 4.4 THE DECREASING DISTANCE PHENOMENON

Interestingly the estimated distance between images can even decrease when the noise increases. One can see an example in Figure 7 where for $p > 1$ we get a "dip" in the distance, showing that the images are getting closer together, similarly to the "two square images" in Figure 4.

This phenomenon, which at first sight might be surprising, can be explained by the fact that for sparse pictures, the noise appearing between two structures can be used to "bridge" the transport distance between them, like we see in Figure 8. Instead of having to transport the mass far away, a large part of it is mapped to surrounding noise, this noise is matched with noise a bit further and so on until all the mass is matched.

### 5 CONCLUSION AND FUTURE WORK

In this paper, we have investigated the behavior of the signed Wasserstein distance under noise corruption of the pictures. Our theoretical contributions provide bounds for various situations of interest. In particular, certain bounds establish a better noise robustness of the signed Wasserstein

distance than the ubiquitous $L^2$ metric. Our numerical experiments on the DOTMark dataset corroborate these findings, with empirical results confirming that the $W_2$ distance is more resilient to noise than both $L^2$ and $W_1$ distances. These results make a strong case for its use in noise-plagued applications like cryo-EM.

Despite these results, there remains venue for additional work. A primary challenge would be to establish a sharp bounds for $\mathbb{E}W_p^{\pm}(\mu_\varepsilon, \nu_\varepsilon) - \mathbb{E}W_p^{\pm}(\mu, \nu)$, which is hindered by the lack of triangle inequality. The numerical experiments further suggest that our bounds, despite capturing the correct behavior, are not tight. Finally, as our theory suggests that robustness increases with higher values of $p$, the interest of such choices of exponents for practical applications should be investigated in future works.

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
