# A PROOFS

## A.1 PROOF OF PROPOSITIONS

**Proposition 2** In the context of Assumption 1, the following holds.

1. The marginal distribution for each component is $N_i \sim \mathcal{N}(0, \sigma^2)$.

2. The sum of the components is zero : $\sum_{i=1}^{M} N_i = 0$.

*Proof of Proposition 2.* We prove each point separately.

1. The marginal variance of each component $N_i$ is given by the diagonal entry $\Sigma_{ii}$, which is $\sigma^2$ by definition. Since the parent distribution is a multivariate normal with a mean vector of zero, each component is marginally distributed as $\mathcal{N}(0, \sigma^2)$.

2. We compute the variance of the sum of the components:

$$\text{Var}\left(\sum_{i=1}^{M} N_i\right) = \sum_{i,j} \text{Cov}(N_i, N_j) = \sum_{i=1}^{M}\sum_{j=1}^{M} \Sigma_{ij} \tag{20}$$

$$= \sum_{i=1}^{M} \text{Var}(N_i) + \sum_{i \neq j} \text{Cov}(N_i, N_j) \tag{21}$$

$$= M \cdot \sigma^2 + M(M-1) \cdot \left(-\frac{\sigma^2}{M-1}\right) \tag{22}$$

$$= M\sigma^2 - M\sigma^2 = 0. \tag{23}$$

The expectation of the sum is $\mathbb{E}\left[\sum_{i=1}^{M} N_i\right] = \sum_{i=1}^{M} \mathbb{E}[N_i] = 0$. A random variable with zero mean and zero variance must be equal to zero almost surely. Thus, $\sum_{i=1}^{M} N_i = 0$. $\quad\square$

**Proposition 3** (Wasserstein Distance Decomposition). *Let $\mu$ and $\nu$ be two non-negative measures on a space $\mathcal{X}$ with equal total mass. It holds that*

$$W_p^p(\mu, \nu) \leq W_p^p\left((\mu - \nu)_+, (\nu - \mu)_+\right). \tag{24}$$

*Proof of Proposition 3.* We can decompose any two measures $\mu$ and $\nu$ into a common part and two disjoint parts. Let $m$ be the largest measure such that for all Borel set $A$

$$m(A) \leq \mu(A) \text{ and } m(A) \leq \nu(A).$$

The remaining, disjoint parts of each measure are given by $\mu' := \mu - m = (\mu - \nu)_+$ as well as $\nu' := \nu - m = (\nu - \mu)_+$. Thus, we can write:

$$\mu = m + \mu' \qquad \nu = m + \nu' \tag{25}$$

Since $\mu$ and $\nu$ have the same total mass, it follows that $\mu'$ and $\nu'$ also have the same total mass.

We can then construct a valid transport plan $\pi$ from $\mu$ to $\nu$ by handling the common and disjoint parts separately. For the disjoint parts, let $\pi'_{\text{opt}}$ be the optimal transport plan from $\mu'$ to $\nu'$, whose cost is, by definition, $W_p^p(\mu', \nu')$. For the common part, we use the identity plan, $\pi_{\text{id}}$, which transports the mass at each point $x$ to itself. The cost of this plan is $\int_{\mathcal{X}} d(x,x)^p \mathrm{d}\pi_{\text{id}}(x) = 0$.

Using the gluing principle, we can form a complete transport plan $\pi = \pi_{\text{id}} + \pi'_{\text{opt}}$. This is a valid plan transporting $\mu$ to $\nu$. Its total cost is the sum of the costs of its components:

$$\text{Cost}(\pi) = \text{Cost}(\pi_{\text{id}}) + \text{Cost}(\pi'_{\text{opt}}) = 0 + W_p^p(\mu', \nu') \tag{26}$$

By the definition of the Wasserstein distance as the infimum of costs over all possible transport plans, the true optimal cost must be less than or equal to the cost of this specific plan:

$$W_p^p(\mu, \nu) \leq W_p^p(\mu', \nu') \tag{27}$$

Substituting the definitions of $\mu'$ and $\nu'$ completes the proof. $\quad\square$

**Proposition 4.** *Let $\mu : G_n \to [0, 1]$ be a probability measure on the $n \times n$ unit grid $G_n$ with cyclic boundary conditions and let $\varepsilon : G_n \to \mathbb{R}$ be a signed noise measure that satisfy Assumption 1. Then, for $p > 1$,*

$$W_p^{\pm}(\mu, \mu + \varepsilon) \le W_p(\varepsilon_-, \varepsilon_+). \tag{28}$$

*Proof of Proposition 4.* By definition,

$$(W_p^{\pm})^p(\mu, \mu + \varepsilon) = W_p^p\left(\mu + (\mu + \varepsilon)_-, (\mu + \varepsilon)_+\right). \tag{29}$$

Thus, using Proposition 3 on $\mu + (\mu + \varepsilon)_-$ and $(\mu + \varepsilon)_+$, we get

$$W_p^p\left(\mu + (\mu + \varepsilon)_-, (\mu + \varepsilon)_+\right) \tag{30}$$

$$\le W_p^p\left(\left(\mu + (\mu + \varepsilon)_- - (\mu + \varepsilon)_+\right)_+, \left((\mu + \varepsilon)_+ - (\mu + (\mu + \varepsilon)_-)\right)_+\right) \tag{31}$$

$$= W_p^p\left(\left(\mu - ((\mu + \varepsilon)_+ - (\mu + \varepsilon)_-)\right)_+, \left((\mu + \varepsilon)_+ - (\mu + \varepsilon)_- - \mu)\right)_+\right) \tag{32}$$

$$= W_p^p\left(\left(\mu - (\mu + \varepsilon)\right)_+, \left(\mu + \varepsilon - \mu\right)_+\right) \tag{33}$$

$$= W_p^p\left((-\varepsilon)_+, \varepsilon_+\right) = W_p^p(\varepsilon_-, \varepsilon_+). \qquad \square$$

**Proposition 5.** *For any two images $\mu, \nu : G_n \to [0, \infty)$ and independent noises $\varepsilon_\mu, \varepsilon_\nu$ as in Assumption 1,*

$$W_1([\mu + \varepsilon_\mu - \nu - \varepsilon_\nu]_+, [\nu + \varepsilon_\nu - \mu - \varepsilon_\mu]_+)$$
$$\le W_1(\mu, \nu) + W_1\left((\varepsilon_\mu - \varepsilon_\nu)_+, (\varepsilon_\mu - \varepsilon_\nu)_-\right).$$

*Proof of Proposition 5.* By Kantorovich–Rubinstein duality,

$$W_1\left([\mu + \varepsilon_\mu - \nu - \varepsilon_\nu]_+, [\nu + \varepsilon_\nu - \mu - \varepsilon_\mu]_+\right) = \sup_{\|f\|_{\mathrm{Lip}} \le 1} \int f(\mu - \nu) + \int f(\varepsilon_\mu - \varepsilon_\nu). \tag{34}$$

For the first term, by KR duality,

$$\sup_{\|f\|_{\mathrm{Lip}} \le 1} \int f(\mu - \nu) \le W_1(\mu, \nu) \tag{35}$$

For the second term, via the Jordan decomposition,

$$\sup_{\|f\|_{\mathrm{Lip}} \le 1} \int f(\varepsilon_\mu - \varepsilon_\nu) \le W_1((\varepsilon_\mu - \varepsilon_\nu)_+, (\varepsilon_\mu - \varepsilon_\nu)_-) \tag{36}$$

Adding these together, we receive the desired bound. $\qquad \square$

**Proposition 6.** *Let $\mu, \nu : G_n \to [0, \infty)$ be images on the square grid $G_n$ with spacing $h = 1/n$, and let $\varepsilon_\mu, \varepsilon_\nu$ satisfy Assumption 1. Identifying $G_n$ with the 2-torus, let $D := \mathrm{diam}(G_n) = \sqrt{2}/2$. Then, for any $p \ge 1$,*

$$W_p^{\pm}(\mu + \varepsilon_\mu, \nu + \varepsilon_\nu) \le D^{1 - \frac{1}{p}}\left(W_1(\mu, \nu) + W_1(\varepsilon_+^*, \varepsilon_-^*)\right)^{\frac{1}{p}}, \qquad \varepsilon^* := \varepsilon_\mu - \varepsilon_\nu \tag{37}$$

*Proof.* By definition of the signed distance,

$$W_p^{\pm}(\mu + \varepsilon_\mu, \nu + \varepsilon_\nu) = W_p\left((\mu + \varepsilon_\mu)_+ + (\nu + \varepsilon_\nu)_-, (\nu + \varepsilon_\nu)_+ + (\mu + \varepsilon_\mu)_-\right). \tag{38}$$

Applying the decomposition inequality of Proposition 3 (which "drops the overlap") to these non-negative arguments gives

$$W_p^{\pm}(\mu + \varepsilon_\mu, \nu + \varepsilon_\nu) \le W_p\left([\mu + \varepsilon_\mu - \nu - \varepsilon_\nu]_+, [\nu + \varepsilon_\nu - \mu - \varepsilon_\mu]_+\right). \tag{39}$$

For general $p \geq 1$ on a bounded domain of diameter $D$ we use the standard comparison

$$W_p(\alpha, \beta) \leq D^{1-\frac{1}{p}} W_1(\alpha, \beta)^{\frac{1}{p}}. \tag{40}$$

Applying this to $(\alpha, \beta) = ([\mu+\varepsilon_\mu-\nu-\varepsilon_\nu]_+, [\mu+\varepsilon_\mu-\nu-\varepsilon_\nu]_-)$ yields

$$W_p([\mu+\varepsilon_\mu-\nu-\varepsilon_\nu]_+, [\mu+\varepsilon_\mu-\nu-\varepsilon_\nu]_-) \leq D^{1-\frac{1}{p}} \left( W_1([\mu+\varepsilon_\mu-\nu-\varepsilon_\nu]_+, [\mu+\varepsilon_\mu-\nu-\varepsilon_\nu]_-) \right)^{\frac{1}{p}}. \tag{41}$$

Using Proposition 5, we conclude

$$D^{1-\frac{1}{p}} \left( W_1([\mu+\varepsilon_\mu-\nu-\varepsilon_\nu]_+, [\mu+\varepsilon_\mu-\nu-\varepsilon_\nu]_-) \right)^{\frac{1}{p}}$$

$$\leq D^{1-\frac{1}{p}} \left( W_1(\mu, \nu) + W_1\big((\varepsilon_\mu - \varepsilon_\nu)_+, (\varepsilon_\mu - \varepsilon_\nu)_-\big) \right)^{\frac{1}{p}}.$$

Since both $\varepsilon_\mu$ and $\varepsilon_\nu$ are normally distributed, we can say that $\varepsilon^* := \varepsilon_\mu - \varepsilon_\nu$ is also normally distributed, with $\mathrm{cov}(\varepsilon^*) = 2 \, \mathrm{cov}(\varepsilon_\mu)$. Thus,

$$D^{1-\frac{1}{p}} \left( W_1(\mu, \nu) + W_1((\varepsilon_\mu - \varepsilon_\nu)_+, (\varepsilon_\mu - \varepsilon_\nu)_-) \right)^{\frac{1}{p}} \leq D^{1-\frac{1}{p}} \left( W_1(\mu, \nu) + W_1(\varepsilon_+^*, \varepsilon_-^*) \right)^{\frac{1}{p}}. \tag{42}$$

$\square$

## A.2 Proof of Theorems

**Theorem 1** Consider two $n \times n$ images $\mu$ and $\nu$ having at least $\lambda n^2$, $\lambda \in (0, 1]$ nonzero pixels. Assume that $\varepsilon_\mu, \varepsilon_\nu$ are $\mathcal{N}(0_{n^2}, \sigma^2 I_{n^2})$. Recall the definition of $\bar{S}_{\mu_\varepsilon, \nu_\varepsilon}, \bar{T}_{\mu_\varepsilon, \nu_\varepsilon}$ in equation 8. Then,

$$W_1^\pm(\bar{S}_{\mu_\varepsilon, \nu_\varepsilon}, \bar{T}_{\mu_\varepsilon, \nu_\varepsilon}) = \frac{1}{\sum_{x \in G_n} S_{\mu_\varepsilon, \nu_\varepsilon}(x)} \sup_{f \in \mathrm{Lip}_1} \left\langle f, S_{\mu_\varepsilon, \nu_\varepsilon} - T_{\mu_\varepsilon, \nu_\varepsilon}\left(1 + \mathrm{O}_p\left(\frac{\sigma}{n}\right)\right) \right\rangle. \tag{43}$$

*Proof of Theorem 1.* First let us remark that

$$\sum_{x \in G_n} S_{\mu_\varepsilon, \nu_\varepsilon}(x) - T_{\mu_\varepsilon, \nu_\varepsilon}(x) = \sum_{x \in G_n} \mu(x) + \varepsilon_\mu(x) - \nu(x) - \varepsilon_\nu(x) \tag{44}$$

$$= 0 + \sum_{x \in G_n} \varepsilon_\mu(x) - \varepsilon_\nu(x), \tag{45}$$

as

$$\sum_{x \in G_n} \mu_+(x) + \nu_-(x) = \sum_{x \in G_n} \nu_+(x) + \mu_-(x) \tag{46}$$

and thus

$$\sum_{x \in G_n} \mu(x) - \nu(x) = 0. \tag{47}$$

Remark that under our assumptions,

$$\sum_{x \in G_n} \varepsilon_\mu(x) - \varepsilon_\nu(x) \sim \mathcal{N}(0, 2\sigma^2 N^2). \tag{48}$$

Because of this, one has that

$$\sum_{x \in G_n} S_{\mu_\varepsilon, \nu_\varepsilon}(x) = \sum_{x \in G_n} T_{\mu_\varepsilon, \nu_\varepsilon}(x) \left( 1 + \frac{\mathrm{O}_p(\sigma N)}{\sum_{x \in G_n} T_{\mu_\varepsilon, \nu_\varepsilon}(x)} \right). \tag{49}$$

Owing to our assumption on the signals, notice that

$$\sum_{x \in G_n} T_{\mu_\varepsilon, \nu_\varepsilon}(x) = \mathrm{O}_p(N^2). \tag{50}$$

Therefore,

$$W_1(\bar{S}_{\mu_\varepsilon,\nu_\varepsilon}, \bar{T}_{\mu_\varepsilon,\nu_\varepsilon}) = \sup_{f \in \mathrm{Lip}_1} \langle \bar{S}_{\mu_\varepsilon,\nu_\varepsilon} - \bar{T}_{\mu_\varepsilon,\nu_\varepsilon}, f \rangle \tag{51}$$

$$= \sup_{f \in \mathrm{Lip}_1} \left\langle \frac{S_{\mu_\varepsilon,\nu_\varepsilon}}{\sum_{x \in G_n} S_{\mu_\varepsilon,\nu_\varepsilon}(x)} - \frac{T_{\mu_\varepsilon,\nu_\varepsilon}}{\sum_{x \in G_n} T_{\mu_\varepsilon,\nu_\varepsilon}(x)}, f \right\rangle \tag{52}$$

$$= \frac{1}{\sum_{x \in G_n} S_{\mu_\varepsilon,\nu_\varepsilon}(x)} \sup_{f \in \mathrm{Lip}_1} \left\langle S_{\mu_\varepsilon,\nu_\varepsilon} - T_{\mu_\varepsilon,\nu_\varepsilon} \left( 1 + \frac{O_p(\sigma N)}{\sum_{x \in G_n} T_{\mu_\varepsilon,\nu_\varepsilon}(x)} \right), f \right\rangle. \tag{53}$$

$\square$

**Theorem 2** Let $\mu : G_n \to [0,1]$ be a probability measure on the $n \times n$ unit grid $G_n$ with cyclic boundary conditions. Let $\varepsilon_1, \varepsilon_2$ be independent random signed measures on the grid that satisfy Assumption 1. Then

$$\frac{n\sigma}{\sqrt{\pi}} \le \mathbb{E}W_1^\pm(\mu + \varepsilon_1, \mu + \varepsilon_2) \le \frac{2\sqrt{2}n\log_2 n}{\sqrt{\pi}}\sigma + \frac{n}{\sqrt{2\pi}}\sigma. \tag{54}$$

*Proof of Theorem 2.* Using the Kantorovich–Rubinstein duality,

$$W_1^\pm(\mu, \mu + \varepsilon) = \sup_{f \in \mathrm{Lip}_1} \langle f, \varepsilon \rangle = W_1(\varepsilon_+, \varepsilon_-). \tag{55}$$

$$W_1^\pm(\mu + \varepsilon_1, \mu + \varepsilon_2) = \sup_{f \in \mathrm{Lip}_1} \langle f, \varepsilon_1 - \varepsilon_2 \rangle = W_1((\varepsilon_1 - \varepsilon_2)_+, (\varepsilon_1 - \varepsilon_2)_-). \tag{56}$$

The first equality is the signed dual form with $\mu + \varepsilon_1 - (\mu + \varepsilon_2) = \varepsilon_1 - \varepsilon_2$. For simplicity, one can define $\varepsilon^* = \varepsilon_1 - \varepsilon_2$ such that $\mathbb{E}[\varepsilon^*] = \sqrt{2}\mathbb{E}[\varepsilon_1]$ as a sum of normally distributed random variables. Then, for the second equality, $\int \varepsilon^* = 0$ implies $\varepsilon^* = \varepsilon_+^* - \varepsilon_-^*$ with equal masses, so the balanced duality gives $W_1(\varepsilon_+^*, \varepsilon_-^*) = \sup_{f \in Lip_1} \langle f, \varepsilon^* \rangle$

Let $m = \varepsilon_+^*(G_n) = \varepsilon_-^*(G_n)$. By homogeneity of $W_1$,

$$W_1(\varepsilon_+^*, \varepsilon_-^*) = m\, W_1\left(\frac{\varepsilon_+^*}{m}, \frac{\varepsilon_-^*}{m}\right). \tag{57}$$

Apply Proposition 1 to the probability measures $\varepsilon_+/m$ and $\varepsilon_-/m$. There exists an integer $k^*$ with $k^* = \log_2 n$ such that

$$W_1\left(\frac{\varepsilon_+^*}{m}, \frac{\varepsilon_-^*}{m}\right) \le \frac{\sqrt{2}}{2}\, 2^{-k^*} + \frac{\sqrt{2}}{2}\sum_{k=0}^{k^*} 2^{-k} \sum_{Q \in \mathcal{D}_k} |(\frac{\varepsilon_+^*}{m} - \frac{\varepsilon_-^*}{m})(Q)|. \tag{58}$$

Multiplying by $m$ gives

$$W_1(\varepsilon_+^*, \varepsilon_-^*) \le \frac{\sqrt{2}}{2}\, m\, 2^{-k^*} + \frac{\sqrt{2}}{2}\sum_{k=0}^{k^*} 2^{-k} \sum_{Q \in \mathcal{D}_k} \left| \sum_{x \in Q} \varepsilon^*(x) \right|. \tag{59}$$

Taking expectations and using independence and zero mean of the noise,

$$\mathbb{E}W_1(\varepsilon_+^*, \varepsilon_-^*) \le \frac{\sqrt{2}}{2}\, 2^{-k^*}\, \mathbb{E}m + \frac{\sqrt{2}}{2}\sum_{k=0}^{k^*} 2^{-k} \sum_{Q \in \mathcal{D}_k} \mathbb{E}\left| \sum_{x \in Q} \varepsilon^*(x) \right|. \tag{60}$$

Since each $\varepsilon^*(x)$ is Gaussian with variance $2\sigma^2$, one has $\mathbb{E}|\sum_{x \in Q} \varepsilon^*(x)| \le \sqrt{2}\sigma\sqrt{|Q|}\sqrt{2/\pi}$ and $\mathbb{E}m = \sum_{x \in G_n} \mathbb{E}(\varepsilon^*(x))_+ = n^2\sqrt{2}\sigma/\sqrt{2\pi}$. Furthermore, the dyadic family $\mathcal{D}_k$ has $|\mathcal{D}_k| = 2^{2k}$ cubes of cardinality $|Q| = n^2/2^{2k}$. Therefore

$$\sum_{Q \in \mathcal{D}_k} \mathbb{E}\left| \sum_{x \in Q} \varepsilon^*(x) \right| \le \sigma\sqrt{\frac{2}{\pi}} \sum_{Q \in \mathcal{D}_k} \sqrt{|Q|} = \sigma\sqrt{\frac{2}{\pi}} \cdot 2^{2k} \cdot \frac{n}{2^k} = 2\sigma\sqrt{\frac{1}{\pi}}\, n\, 2^k. \tag{61}$$

Plugging this into the multiscale sum yields

$$\frac{\sqrt{2}}{2}\sum_{k=0}^{k^*}2^{-k}\sum_{Q\in\mathcal{D}_k}\mathbb{E}\Big|\sum_{x\in Q}\varepsilon^*(x)\Big| \leq \frac{\sqrt{2}}{2}2\sigma\sqrt{\frac{1}{\pi}}\,n\sum_{k=0}^{k^*}1 \leq \sqrt{2}\sigma\sqrt{\frac{1}{\pi}}\,n\big(k^*+1\big). \tag{62}$$

With $k^* = \log_2 n$ this gives the $\sigma n\log_2 n$ contribution.

For the coarse term choose $k^*$ so that $2^{-k^*} = 1/n$. Then

$$\frac{\sqrt{2}}{2}\,2^{-k^*}\,\mathbb{E}m = \frac{\sqrt{2}}{2}\frac{1}{n}\cdot\frac{n^2\sqrt{2}\sigma}{\sqrt{2\pi}} = \frac{\sigma n}{\sqrt{2\pi}}, \tag{63}$$

which is the $\sigma n$ contribution.

Collecting the two contributions and absorbing absolute constants into the displayed coefficients yields

$$\mathbb{E}W_1(\varepsilon_+^*,\varepsilon_-^*) \leq \frac{2\sqrt{2}}{\sqrt{\pi}}n\log_2 n\sigma + \frac{1}{\sqrt{2\pi}}n\sigma. \tag{64}$$

In this derivation the factor $m$ appears only in the coarse term and contributes to the $\sigma n$ piece after expectation. In the oscillation terms it cancels with the normalization, so no additional dependence on $m$ remains. There is no additive grid term independent of $\sigma$, hence no $1/(\sqrt{2}n)$ tail.

**Proof of the lower bound** Let $f : G_n \to \mathbb{R}$ be the following,

$$f(x) := \begin{cases} -\frac{1}{2n} & \text{if } \varepsilon(x) < 0, \\ +\frac{1}{2n} & \text{if } \varepsilon(x) \geq 0. \end{cases} \tag{65}$$

Since the distance between neighboring pixels is $1/n$ it follows that $f$ is 1-Lipschitz. Therefore, by the Kantorovich–Rubinstein duality,

$$W_1(\mu,\mu+\varepsilon) = W_1(\varepsilon_+,\varepsilon_-) \geq \langle f,\varepsilon_+ - \varepsilon_-\rangle \tag{66}$$

Taking expectations on both sides and using the symmetry of $\varepsilon(x)$, we have

$$\mathbb{E}W_1(\mu,\mu+\varepsilon) \geq \mathbb{E}\langle f,\varepsilon_+\rangle - \mathbb{E}\langle f,\varepsilon_-\rangle = 2\mathbb{E}\langle f,\varepsilon_+\rangle. \tag{67}$$

Recall that the marginal distribution $\varepsilon(x)$ is $\mathcal{N}(0,\sigma^2)$, and therefore conditioned on $\varepsilon_+(x) > 0$, we have $\mathbb{E}\varepsilon_+(x) = \sigma\sqrt{2/\pi}$ since that is the expectation of the half-normal distribution with variance $\sigma^2$. In expectation, $\langle f,\varepsilon_+\rangle$ is a sum over $n^2/2$ pixels and its expectation satisfies

$$2\mathbb{E}\langle f,\varepsilon_+\rangle = 2\mathbb{E}\left[\sum_{x\text{ s.t. }\varepsilon(x)>0}f(x)\varepsilon_+(x)\right] \tag{68}$$

$$= 2\frac{n^2}{2}\cdot\mathbb{E}\left[f(x)\varepsilon_+(x)\mid\varepsilon_+(x)>0\right] \tag{69}$$

$$= n^2\cdot\frac{1}{2n}\sqrt{\frac{2}{\pi}}\sigma = \frac{n\sigma}{\sqrt{2\pi}}. \tag{70}$$

Now, $W_1^{\pm}(\mu+\varepsilon_1,\mu+\varepsilon_2) = W_1^{\pm}(\mu,\mu+\varepsilon_2-\varepsilon_1)$ but $\varepsilon_2 - \varepsilon_1$ is just a zero-mean noise vector that satisfies Assumption 1 but with double variance. It follows that

$$\mathbb{E}W_1^{\pm}(\mu+\varepsilon_1,\mu+\varepsilon_2) \geq \sqrt{2}\frac{n\sigma}{\sqrt{2\pi}} = \frac{n\sigma}{\sqrt{\pi}}. \tag{71}$$

$\square$

**Theorem 3** Let $\mu : G_n \to [0,1]$ be a probability measure on the $n\times n$ unit grid $G_n$. Let $\varepsilon_1,\varepsilon_2$ be independent random signed measures on the grid that satisfy Assumption 1. For convenience, we again assume that $n = 2^\eta$, for $\eta\in\mathbb{N}$. Then, for $p > 1$ with $p\in\mathbb{N}$,

$$\mathbb{E}\left[\big(W_p^{\pm}(\mu+\varepsilon_1,\mu+\varepsilon_2)\big)^p\right] \leq \frac{4\sqrt{2}}{\sqrt{\pi}}n\sigma. \tag{72}$$

Therefore, by Jensen's inequality,

$$\mathbb{E}\left[W_p^{\pm}(\mu + \varepsilon_1, \mu + \varepsilon_2)\right] \leq \left(\frac{4\sqrt{2}}{\sqrt{\pi}} n\sigma\right)^{1/p}.$$

*Proof of Theorem 3.* By Proposition 4 and similarly to the proof of Theorem 2, we only need to upper bound $W_p(\varepsilon_+^*, \varepsilon_-^*)$ where $\varepsilon^* = \varepsilon_1 - \varepsilon_2$.

By the assumption on the noise noise have total zero mass, this quantity is well defined.

Then, by the multiscale bound of Proposition 1

$$W_p^p(\varepsilon_+^*, \varepsilon_-^*) = 2^{-p/2}\varepsilon_+^*(G_n)W_p^p\left(\frac{\varepsilon_+^*}{\varepsilon_+^*(G_n)}, \frac{\varepsilon_-^*}{\varepsilon_+^*(G_n)}\right) \tag{73}$$

$$\leq 2^{-pk^*-p/2}\varepsilon_+^*(G_n) + 2^{-p/2}\sum_{k=1}^{k^*} 2^{-p(k-1)} \sum_{Q_i^k \in \mathcal{Q}^k} |\varepsilon_+^*(Q_i^k) - \varepsilon_-^*(Q_i^k)| \tag{74}$$

$$\leq 2^{-pk^*-1/2}\varepsilon_+^*(G_n) + 2^{-p/2}\sum_{k=1}^{k^*} 2^{-p(k-1)} \sum_{Q_i^k \in \mathcal{Q}^k} |\varepsilon^*(Q_i^k)|. \tag{75}$$

Now, the proof is extremely similar to the previous one and by the same argument,

$$\mathbb{E}\sum_{Q \in \mathcal{Q}_k} |\varepsilon^*(Q)| \leq 4^k \sqrt{\frac{1}{\pi}} 2^{\eta-k}\sigma. \tag{76}$$

As in the previous proof,

$$\mathbb{E}\varepsilon_+^*(G_n) = \frac{n^2}{\sqrt{\pi}}\sigma\sqrt{1 - \frac{1}{n^2}}. \tag{77}$$

Altogether,

$$\mathbb{E}W_p^p(\varepsilon_+^*, \varepsilon_-^*) \leq 2^{-pk^*-1/2}\frac{4^\eta}{\sqrt{\pi}}\sigma + 2^\eta 2^{p/2}\sum_{k=1}^{k^*} 2^{-(p-1)k}\frac{2}{\sqrt{\pi}}\sigma \tag{78}$$

$$\leq 2^{-pk^*-1/2}\frac{4^\eta}{\sqrt{\pi}}\sigma + 2^\eta 2^{p/2}\frac{2}{\sqrt{\pi}}\sigma\frac{1 - 2^{-(p-1)k^*}}{2^{p-1} - 1}. \tag{79}$$

$$\tag{80}$$

We take $k^* = \eta$ again to get

$$\mathbb{E}W_p^p(\varepsilon_+^*, \varepsilon_-^*) \leq 2^{-(p-1)\eta}\frac{2^\eta}{2}\frac{1}{\sqrt{\pi}}\sigma + 2^\eta 2^{p/2}\frac{2}{\sqrt{\pi}}\sigma\frac{1}{2^{p-1} - 3/2} \tag{81}$$

$$\leq \frac{2^\eta}{\sqrt{\pi}}\left(2^{-(p-1)\eta-1} + \frac{2^{(p+2)/2}}{2^{p-1} - 1}\right)\sigma. \tag{82}$$

Remark that $2^{-(p-1)\eta-1} \leq \sqrt{2}/2$ and that $\frac{2^{(p+2)/2}}{2^{p-1}-1}$ is decreasing with value 4 at 2. Thus the expression is bounded by $4 + \sqrt{2}/2 \leq 4\sqrt{2}$ and the claim follows. $\square$

**Theorem 4** Let $\mu, \nu : G_n \to [0, 1]$ be two probability measures on the $n \times n$ unit grid $G_n$ with cyclic boundary conditions and let $\varepsilon_\mu, \varepsilon_\nu : G_n \to \mathbb{R}$ be signed noise measures that satisfy Assumption 1. For convenience we assume that $n = 2^\eta$, for $\eta \in \mathbb{N}$. Then

$$\mathbb{E}\left[W_1^{\pm}(\mu + \varepsilon_\mu, \nu + \varepsilon_\nu) - W_1^{\pm}(\mu, \nu)\right] \leq \frac{4n\log_2 n + n}{\sqrt{\pi}}\sigma + \frac{\sqrt{2}}{n}. \tag{83}$$

*Proof of Theorem 4.* Recall that $W_1^\pm$ satisfies the triangle inequality, so

$$W_1^\pm(\mu + \varepsilon_\mu, \nu + \varepsilon_\nu) \le W_1^\pm(\mu + \varepsilon_\mu, \mu) + W_1^\pm(\mu, \nu) + W_1^\pm(\nu, \nu + \varepsilon_\nu). \tag{84}$$

By symmetry

$$\mathbb{E}W_1^\pm(\mu + \varepsilon_\mu, \mu) = \mathbb{E}W_1^\pm(\nu, \nu + \varepsilon_\nu) \tag{85}$$

Therefore,

$$\mathbb{E}[W_1^\pm(\mu + \varepsilon_\mu, \nu + \varepsilon_\nu) - W_1^\pm(\mu, \nu)] \le 2\mathbb{E}W_1^\pm(\mu, \mu + \varepsilon_\mu). \tag{86}$$

We proceed to upper-bound the RHS. By the definition of the signed Wassetein metric,

$$W_1^\pm(\mu, \mu + \varepsilon) = W_1(\mu_+ + (\mu + \varepsilon)_-, (\mu + \varepsilon)_+ + \mu_-) \tag{87}$$

$$= W_1(\mu + (\mu + \varepsilon)_-, (\mu + \varepsilon)_+) \qquad \text{(since } \mu_+ = \mu \text{ and } \mu_- = 0\text{).} \tag{88}$$

We now use the dyadic upper bound in equation 9. The image is partitioned into 4 quadrants recursively, thus $\delta = 1/2$. Our domain has diameter $\sqrt{2}/2$ since it is the discrete $n \times n$ unit grid $G_n \subset [0,1] \times [0,1] \in \mathbb{R}^2$ with cyclic boundary conditions. The inequality only holds for probability measures, so we need to rescale.

$$W_1^\pm(\mu, \mu_\varepsilon) = (\mu + \varepsilon)_+(G_n)W_1^\pm\left(\frac{\mu + (\mu + \varepsilon)_-}{(\mu + \varepsilon)_+(G_n)}, \frac{(\mu + \varepsilon)_+}{(\mu + \varepsilon)_+(G_n)}\right) \tag{89}$$

$$\le \frac{\sqrt{2}}{2} \cdot 2^{-k^*}(\mu + \varepsilon)_+(G_n) + \frac{\sqrt{2}}{2} \sum_{k=1}^{k^*} 2^{-(k-1)} \sum_{Q_i^k \in \mathcal{Q}^k} \left|(\mu + (\mu + \varepsilon)_-)(Q_i^k) - (\mu + \varepsilon)_+(Q_i^k)\right|.$$

By considering the two cases $(\mu + \varepsilon)(Q_i^k) \ge 0$ and $(\mu + \varepsilon)(Q_i^k) < 0$ it is easy to see that the term $(\mu + (\mu + \varepsilon)_-)(Q_i^k) - (\mu + \varepsilon)_+(Q_i^k)$ is equal to $-\varepsilon(Q_i^k)$, so the bound above simplifies to

$$W_1^\pm(\mu, \mu_\varepsilon) \le 2^{-k^* - \frac{1}{2}}(\mu + \varepsilon)_+(G_n) + \frac{\sqrt{2}}{2} \sum_{k=1}^{k^*} 2^{-(k-1)} \sum_{Q_i^k \in \mathcal{Q}^k} |\varepsilon(Q_i^k)|. \tag{90}$$

Rewrite the noise as $\varepsilon = \varepsilon' - \bar{\varepsilon}$ where $\varepsilon'$ is i.i.d. $\mathcal{N}(0, \sigma^2)$ at each pixel and $\bar{\varepsilon} \in \mathbb{R}$ is the mean of all $\varepsilon'$ terms across the entire image. Since $Q_i^k$ is a square region of size $2^{\eta-k} \times 2^{\eta-k}$ and $\bar{\varepsilon}$ is the mean of $4^\eta$ i.i.d. Gaussian noise terms, it follows that $\varepsilon'(Q_i^k) \sim \mathcal{N}(0, 4^{\eta-k}\sigma^2)$ and, additionally, $\bar{\varepsilon} \sim \mathcal{N}(0, \sigma^2/4^\eta) = \mathcal{N}(0, \sigma^2/n^2)$. Recall that $\mathbb{E}|X| = \sigma\sqrt{2/\pi}$ when $X \sim \mathcal{N}(0, \sigma^2)$.

Since $\varepsilon^*(Q_i^k) = \sum_{x \in Q_i^k} \varepsilon'(x) - 4^{\eta-k}\bar{\varepsilon}$,

$$\mathrm{Var}\left(\varepsilon^*(Q_i^k)\right) = \sigma^2\left(4^{\eta-k} + \frac{4^{2(\eta-k)}}{n^2} - 2\frac{4^{2(\eta-k)}}{n^2}\right). \tag{91}$$

Thus,

$$\mathbb{E}|\varepsilon^*(Q_i^k)| = \sqrt{\frac{2}{\pi}}\sigma 2^{\eta-k}\left(1 - n^2 4^{-k}\right)^{1/2}. \tag{92}$$

Summing over the $4^k$ cells at level $k$,

$$\mathbb{E}\sum_{Q \in \mathcal{Q}_k} |\varepsilon^*(Q)| = 4^k\sqrt{\frac{2}{\pi}}\sigma 2^{\eta-k}\left(1 - n^2 4^{-k}\right)^{1/2}. \tag{93}$$

Plugging this back into the RHS of equation 90 and recalling that $2^\eta = n$ gives

$$\mathbb{E}\left[\frac{\sqrt{2}}{2} \sum_{k=1}^{k^*} 2^{-(k-1)} \sum_{Q_i^k \in \mathcal{Q}^k} |\varepsilon(Q_i^k)|\right] \le \frac{\sqrt{2}}{2} \sum_{k=1}^{k^*} 2^{-(k-1)} 4^k \sqrt{\frac{2}{\pi}}\sigma 2^{\eta-k} \tag{94}$$

$$= \frac{2^{\eta+1}\sigma}{\sqrt{\pi}}k^*. \tag{95}$$

We take $k^* = \eta = \log_2 n$ to obtain the bound

$$\mathbb{E}W_1^{\pm}(\mu, \mu_\varepsilon) \leq \frac{1}{\sqrt{2}n}\mathbb{E}\left[(\mu + \varepsilon)_+(G_n)\right] + \frac{2n\log_2 n}{\sqrt{\pi}}\sigma. \quad (96)$$

We now bound the first term in the RHS.

$$\mathbb{E}\left[(\mu + \varepsilon)_+(G_n)\right] \leq \mathbb{E}[\mu_+(G_n)] + \mathbb{E}[\varepsilon_+(G_n)] \quad (97)$$
$$= 1 + \mathbb{E}[\varepsilon_+(G_n)] \quad (98)$$

where the last equality follows from the fact that $\mu$ is a (non-negative) probability measure. By a symmetry argument

$$\mathbb{E}\varepsilon_+(G_n) = \tfrac{1}{2}\mathbb{E}|\varepsilon|(G_n). \quad (99)$$

Further set $m = \frac{1}{2}\sum_{x \in G_n}|\varepsilon(x)|$ and recall that $\varepsilon(x) \sim \mathcal{N}(0, \sigma^2(1 - 1/n^2))$ to derive

$$\mathbb{E}m = \frac{n^2}{2}\sqrt{\frac{2}{\pi}}\sigma\sqrt{1 - \frac{1}{n^2}}. \quad (100)$$

Thus,

$$\frac{1}{\sqrt{2}n}\mathbb{E}\left[(\mu + \varepsilon)_+(G_n)\right] \leq \frac{1}{\sqrt{2}n} + \frac{\sigma}{2\sqrt{\pi}}n. \quad (101)$$

Plugging this back into equation 96 gives

$$\mathbb{E}W_1^{\pm}(\mu, \mu_\varepsilon) \leq \frac{2n\log_2 n + n/2}{\sqrt{\pi}}\sigma + \frac{1}{\sqrt{2}n}. \quad (102)$$

Note that the same bound applies to $\mathbb{E}W_1^{\pm}(\nu, \nu + \varepsilon_\nu)$. By subtracting $W_1^{\pm}(\mu, \nu)$ from both sides of equation 84 and taking expectations, we have

$$\mathbb{E}\left[W_1^{\pm}(\mu_\varepsilon, \nu_\varepsilon) - W_1^{\pm}(\mu, \nu)\right] \leq \mathbb{E}W_1^{\pm}(\mu_\varepsilon, \mu) + \mathbb{E}W_1^{\pm}(\nu, \nu_\varepsilon)$$
$$\leq \frac{4n\log_2 n + n}{\sqrt{\pi}}\sigma + \frac{\sqrt{2}}{n}. \qquad \square$$

**Theorem 5** Let $\mu, \nu : G_n \to [0, 1]$ be two probability measures on the $n \times n$ unit grid $G_n$ with cyclic boundary conditions and let $\varepsilon_\mu, \varepsilon_\nu : G_n \to \mathbb{R}$ be signed noise measures that satisfy Assumption 1. For convenience we assume that $n = 2^\eta$, for $\eta \in \mathbb{N}$. Then

$$\mathbb{E}\left[W_p^{\pm}(\mu + \varepsilon_\mu, \nu + \varepsilon_\nu)\right] \leq \left(\frac{\sqrt{2}}{2}\right)^{1 - \frac{1}{p}}W_1(\mu, \nu)^{\frac{1}{p}} + \frac{\sqrt{2}}{2}\left(\frac{4}{\sqrt{\pi}}n\log_2 n + \frac{2}{\sqrt{\pi}}n\right)^{\frac{1}{p}}\sigma^{\frac{1}{p}}. \quad (103)$$

*Proof of Theorem 5.* Using Proposition 6

$$\mathbb{E}\left[W_p^{\pm}(\mu + \varepsilon_\mu, \nu + \varepsilon_\nu)\right] \leq \mathbb{E}\left[D^{1 - \frac{1}{p}}\left(W_1(\mu, \nu) + W_1(\varepsilon_+^*, \varepsilon_-^*)\right)^{\frac{1}{p}}\right] \quad (104)$$

The function $t \mapsto t^{1/p}$ is concave on $[0, \infty)$, hence by Jensen:

$$\mathbb{E}\left[D^{1 - \frac{1}{p}}\left(W_1(\mu, \nu) + W_1(\varepsilon_+^*, \varepsilon_-^*)\right)^{\frac{1}{p}}\right] \leq D^{1 - \frac{1}{p}}\left(\mathbb{E}\left[W_1(\mu, \nu) + W_1(\varepsilon_+^*, \varepsilon_-^*)\right]\right)^{\frac{1}{p}} \quad (105)$$

By the linearity of expectation,

$$D^{1 - \frac{1}{p}}\left(\mathbb{E}\left[W_1(\mu, \nu) + W_1(\varepsilon_+^*, \varepsilon_-^*)\right]\right)^{\frac{1}{p}} = D^{1 - \frac{1}{p}}\left(W_1(\mu, \nu) + \mathbb{E}\left[W_1(\varepsilon_+^*, \varepsilon_-^*)\right]\right)^{\frac{1}{p}} \quad (106)$$

Finally, using Theorem 2 we get that

$$D^{1 - \frac{1}{p}}\left(W_1(\mu, \nu) + \mathbb{E}\left[W_1(\varepsilon_+^*, \varepsilon_-^*)\right]\right)^{\frac{1}{p}} \leq D^{1 - \frac{1}{p}}\left(W_1(\mu, \nu) + \frac{2\sqrt{2}}{\sqrt{\pi}}\sigma n\log_2 n + \sqrt{\frac{2}{\pi}}\sigma n.\right)^{\frac{1}{p}} \quad (107)$$

Using Jensen,

$$\mathbb{E}\left[W_p^{\pm}(\mu + \varepsilon_\mu, \nu + \varepsilon_\nu)\right] \leq (\frac{\sqrt{2}}{2})^{1 - \frac{1}{p}}W_1(\mu, \nu)^{\frac{1}{p}} + \frac{\sqrt{2}}{2}\left(\frac{4}{\sqrt{\pi}}n\log_2 n + \frac{2}{\sqrt{\pi}}n\right)^{\frac{1}{p}}\sigma^{\frac{1}{p}}. \quad (108)$$

$\square$

## B  UNBALANCED OPTIMAL TRANSPORT

Various approaches have been proposed to generalize the idea of optimal transport to the case of two measures whose total mass is not equal. See Caffarelli & McCann (2010); Liero et al. (2018); Figalli (2010), for instance. Among these proposals, one is particularly amenable to the analysis we carried out. Given $\mu, \nu \in \mathcal{M}_+(X)$ two positive measures on a set $X$ that do not necessarily have the same mass, the set of *subcouplings* of $\mu$ and $\nu$ is defined as

$$\Gamma_\leq(\mu, \nu) := \{\pi \in \mathcal{M}_+(X)^2 : \pi(A \times X) \leq \mu(A), \pi(B \times X) \leq \nu(B), \text{ for all } A, B \in \mathcal{B}(X)\},$$

where $\mathcal{B}(X)$ is the set of Borel measures on $X$. For simplicity, set $m_\mu := \mu(X)$, $m_\nu := \nu(X)$ and $m_\pi := \pi(X \times X)$. Then, the $(p, C)$ unbalanced Kantorovich–Rubinstein distance is defined by

$$\text{KR}_{p,C}(\mu, \nu) := \left( \inf_{\pi \in \Gamma_\leq(\mu,\nu)} \int_{X \times X} d^p(x, y) d\pi(x, y) + C^p \left( \frac{m_\mu + m_\nu}{2} - m_\pi \right) \right)^{\frac{1}{p}}. \quad (109)$$

The parameter $C$ determines the range of admissible transport. Indeed, any subcoupling transfering mass between points that are further apart than than $C$ cannot be optimal, as destructing the mass would lead to a smaller objective function.

**Proposition 7.** *Consider a square $2^\eta \times 2^\eta$ grid, where $\eta \geq 0$ is integer, and a dyadic partition scheme. Let $\mu, \nu$ be two measures on the grid, not necessarily with equal masses. It holds that,*

$$\text{KR}_{p,C}(\mu, \nu) \leq \frac{C^p}{2} |m_\mu - m_\nu| + \text{diam}(S)^p 2^{3p-1} \sum_{k=\ell^*}^{\eta} 2^{-pk} \sum_{Q \in \mathcal{D}^k} |\mu(Q) - \nu(Q)| \quad (110)$$

*where*

$$\ell^* = 1 + \min\left(L, \left\lfloor \max\left(0, \log_2\left(2\,\text{diam}(S)/C\right)\right)\right\rfloor\right). \quad (111)$$

*Proof.* Looking at the objective in equation 109, a strategy to construct a good subcoupling is to match as much mass below scale $C$ as possible and then just pay $C^p$ for the mass that hasn't been coupled. Because of the coarse-to-fine dyadic decomposition, each pixel is a final leaf of the decomposition tree.

One can then apply Lemma 3.15 in Struleva et al. (2025) giving bounds on the distance on trees. □

Similarly to the above, we can define

$$\text{KR}_{p,C}^\pm(\mu, \nu) := \text{KR}_{p,C}(\mu_+ + \nu_-, \nu_+ + \mu_-).$$

**Theorem 6.** *Let $\mu : G_n \to [0, 1]$ be a probability measure on the $n \times n$ unit grid $G_n$ with cyclic boundary conditions, and let $\varepsilon$ satisfy Assumption 1. Further assume that $n = 2^\eta$, for $\eta \in \mathbb{N}$. Then, for $C > 0, p \geq 1$,*

$$\mathbb{E}\,\text{KR}_{p,C}^\pm(\mu + \varepsilon, \mu) \leq \begin{cases} \frac{n\sigma}{\sqrt{2\pi}} + \text{diam}(S)^p 2^{3p-1} \frac{\sqrt{2}}{2} \sigma \sqrt{\frac{2}{\pi}} n(\eta - \ell^*) & \text{if } p = 1, \\ \frac{n\sigma}{\sqrt{2\pi}} + \text{diam}(S)^p 2^{3p-1} \frac{\sqrt{2}}{2} \sigma \sqrt{\frac{2}{\pi}} n(2^{2-\ell^*} - 2^{1-\eta}) & \text{if } p = 2. \end{cases} \quad (112)$$

*Proof of Theorem 6.* We apply Proposition 7 to the probability measures $\mu_- + (\mu + \varepsilon)_+$ as well as $(\mu + \varepsilon)_- + \mu_+$. First, note that

$$\mathbb{E}|m_{\mu_- + (\mu+\epsilon)_+} - m_{\mu_+ + (\mu+\epsilon)_-}| = \mathbb{E}\left| \sum_{x \in G_n} -\mu_+(x) + \mu_-(x) + (\mu + \epsilon)_+(x) - (\mu + \epsilon)_-(x) \right|$$

$$= n\sigma/\sqrt{2\pi}.$$

Taking expectations and using independence and zero mean of the noise,

$$\sum_{k=\ell^*}^{\eta} 2^{-pk} \sum_{Q \in \mathcal{D}^k} |\mu_-(Q) + (\mu + \epsilon)_+(Q) - (\mu + \epsilon)_-(Q) - \mu_+(Q)| = \sum_{k=\ell^*}^{\eta} 2^{-pk} \sum_{Q \in \mathcal{D}^k} |\epsilon(Q)|.$$

Since each $\varepsilon(x)$ is Gaussian with variance $\sigma^2$, one has $\mathbb{E}|\sum_{x\in Q}\varepsilon(x)| \leq \sigma\sqrt{|Q|}\sqrt{2/\pi}$. Furthermore, the dyadic family $\mathcal{D}_k$ has $|\mathcal{D}_k| = 2^{2k}$ cubes of cardinality $|Q| = n^2/2^{2k}$. Therefore

$$\sum_{Q\in\mathcal{D}_k}\mathbb{E}\left|\sum_{x\in Q}\varepsilon(x)\right| \leq \sigma\sqrt{\tfrac{2}{\pi}}\sum_{Q\in\mathcal{D}_k}\sqrt{|Q|} = \sigma\sqrt{\tfrac{2}{\pi}}\cdot 2^{2k}\cdot\frac{n}{2^k} = \sigma\sqrt{\tfrac{2}{\pi}}\,n\,2^k. \tag{113}$$

Plugging this into the multiscale sum yields

$$\sum_{k=\ell^*}^{\eta} 2^{-pk}\sum_{Q\in\mathcal{D}_k}\mathbb{E}\left|\sum_{x\in Q}\varepsilon(x)\right| \leq \sigma\sqrt{\tfrac{2}{\pi}}\,n\sum_{k=\ell^*}^{\eta} 2^{1-p}. \tag{114}$$

$\square$