# OpenReview forum: "Quantifying the noise sensitivity of the Wasserstein metric for images"
_ICLR.cc/2026/Conference — Submitted to ICLR 2026_

### Official Review · Reviewer_oX7e · 2025-10-28

**Soundness:** 3
**Presentation:** 3
**Contribution:** 3
**Rating:** 8
**Confidence:** 3

**Summary:**

This paper investigates the sensitivity of the Wasserstein distance to additive pixel-wise noise, providing exact, non-asymptotic bounds and theoretical insights into how noise affects the (signed) 2-Wasserstein distance. The authors show that the error scales with the square root of the noise standard deviation, while the norm scales linearly. They further validate their findings through a range of synthetic and real-data experiments, including a compelling case study on cryo-electron microscopy (cryo-EM) images.
This is a well-executed and thoughtful paper that deepens our understanding of Wasserstein metrics under noise. Its clarity, empirical grounding, and reproducibility make it a strong contribution, even if its focus is somewhat specialized.

**Strengths:**

- The paper is clearly written and the theoretical contributions (and limitations) are made clear.
- Interesting empirical experiments add and support the theoretical contributions nicely.
- The paper is well-motivated by application since it focusses on applications in cyro-EM, a very active research field.
	-  In particular, Figure 6 clearly outlines the usefulness of Wasserstein distances for cyro-EM
- All necessary code is in the supplementary material and is very accessible. Given that I have to review too many conference submission about new algorithms without code, I really appreciate that the authors took the effort to prepare and attach the code for a theory paper.

**Weaknesses:**

- Lines 39-47: I do not feel comfortable with some of the statements here given more recent research over the last five years. While Wasserstein GANs (WGANs) and autoencoders (WAEs) are inspired by OT and are important milestones in generative models, a number of papers have shown that the link between OT and these models is rather weak, e.g. (Stanczuk et al., 2021). Wasserstein GANs profit from clipping weights and controlling Lipschitz constants, whereas Wasserstein autoencoders are straghtforward models that make sense without any knowledge about Wasserstein distances. In addition to WGANs and WAEs, I would also consider OT-based flow matching as a very important milestone of OT-based generative models, see (Lipman et al., 2023; Albertgo et., 2023) for general flow matching background and (Tong et al., 2024; Chemseddine et al., 2025; Mousavi-Hosseini et al., 2025 and many more) for OT-based flow matching.

- This might be more of a ‘niche’ topic at ICLR. Nevertheless, this type of research certainly has its place.

- The proofs are interesting and added to my understanding, but switching between the statements in the main document and the proofs in the supplementary material made comprehension more difficult. I think that the supplementary proof section could benefit from repeating the statements, some additional explanations between proofs and one or two proof sketches.

- I found Figure 2 confusing because it is hard to see the differences between the lines. I would advise to consider two (fits vs. theoretical) or three  (W1 vs W2 vs L2)  side-by-side plots (with equalized axes) and to add legends for ‘fit vs. theoretical’. I think that would it make it easier to look at it.

- Similarly, I would add (fits vs. theoretical) legends to Figure 3/7. There are six lines, but one has to read the caption and the legend together to understand the meaning of each line.

- Figure 4: I would advise a larger font for the x-/y-ticks and labels and thicker lines.

- Line 648: ‘proposition 5’ should have capital ‘P’

- $\mathbb{E}m$ looks weird (line 689) – You could probably drop the expectation here, I guess.

**Questions:**

- My understanding is that all measures/images live on 2-Tours ($G_n$). On the other hand, your noise lives $\mathbb{R}^d$ and during various proof steps you implicitly employ that the measures live on an Euclidean space. While one can certainly extend a measure on a compact domain to the whole $\mathbb{R}^d$ via cyclic conditions, the resulting measure would not necessarily be in $P_2(\mathbb{R}^d)$. Could you please comment on the way this works, respectively, elaborate?

- “For convenience, we again assume that n is a power-of-two.” Please specify in the main text– do you mean $n = 2^m$? Please explain why this matters in the main text. After looking through the proofs, it becomes more tangible, but it is unclear without consultation of the supplementary material.

- What is the meaning of ’≍’ (line 677, 702)?

- The authors consider a rather particular noise model that circumvents the ‘rescaling’ problem. While it is a reasonable choice, I would think that it is more intuitive to consider an unbalanced Wasserstein divergence instead. This would also allow the authors to assume that the images $\mu, ¸\nu$ have different masses. Please comment on the decision not to consider unbalanced optimal transport.

- I am no expert in cryo-EM, but I would be interested in a discussion and maybe an empirical comparison with ‘learned perceptual metrics’, e.g., LPIPS and FID. This would have been a nice addition to the paper. Please comment on the relevance of such neural-network-based metrics.

---

> ### Author Response · Authors · 2025-11-24
> **Answer to the weaknesses section of the review**
>
> We thank the reviewer for their thoughtful critique. We made several improvements and fixes to the paper in light of the review.
>
> A point-by-point response follows:
>
> * _Lines 39-47: While Wasserstein GANs (WGANs) and autoencoders (WAEs) are inspired by OT and are important milestones in generative models, a number of papers have shown that the link between OT and these models is rather weak [...] I would also consider OT-based flow matching as a very important milestone of OT-based generative models [...]_
>
> Thank you for bringing these papers on flow-matching/OT into our attention. We have updated the Related Work part of the introduction to put more emphasis on flow matching and less on WGAN/WAE, since they are inspired by OT but the connection is turned out to be rather weak.
>
> * _the supplementary proof section could benefit from repeating the statements, some additional explanations between proofs and one or two proof sketches._
>
> In the revised manuscript, we have added the theorem statements to the main text directly before the proofs in the Appendix. A few elements sketching the proof of Theorems 2, 3, 4 and  5 have been added in the main text.
>
> * _I found Figure 2 confusing because it is hard to see the differences between the lines. I would advise to consider two (fits vs. theoretical) or three (W1 vs W2 vs L2) side-by-side plots (with equalized axes) and to add legends for ‘fit vs. theoretical’. I think that would it make it easier to look at it._
> * _Similarly, I would add (fits vs. theoretical) legends to Figure 3/7. There are six lines, but one has to read the caption and the legend together to understand the meaning of each line._
> * _Figure 4: I would advise a larger font for the x-/y-ticks and labels and thicker lines._
>
> The above three points were considered and adjusted according to your comments.
>
> * _looks weird (line 689) – You could probably drop the expectation here, I guess._
>
> This is now in line 856. Note that $m$ is the total mass. It is a random variable, so the expectation cannot be dropped.

---

> ### Author Response · Authors · 2025-11-24
> **Answer to the questions of the reviewer**
>
> * _My understanding is that all measures/images live on 2-Torus $G_n$. On the other hand, your noise lives $\mathbb{R}^d$ [...] Could you please comment on the way this works, respectively, elaborate?_
>
> Our images live on an $n \times n$ grid, so each image can be naturally associated with a vector in $\mathbb{R}^{n^2}$. Our noise model involves adding Gaussian i.i.d. variables  to every pixel and then subtracting the mean of the noise values. We have edited Section 3.2 to make this clearer.
>
> As for the 2-torus, we simply set the ground-cost (the distance between pixels) to be cyclic in the experiments and theory. This is done mainly to avoid boundary effects in the experimental results, but is not an important element of our analysis, i.e.,  very similar results with a Euclidean non-cyclic ground cost are easily deduced from our work.
>
> * _Please specify in the main text– do you mean $n=2^m$? Please explain why this matters in the main text._
>
> Thank you for pointing this out. This has been made precise in the main text at each instance.
>
> * _What is the meaning of ’≍’ (line 677, 702)?_
>
> This notation has been removed so that we do not even need to introduce it now.
>
> * _The authors consider a rather particular noise model that circumvents the ‘rescaling’ problem. While it is a reasonable choice, I would think that it is more intuitive to consider an unbalanced Wasserstein divergence instead._
>
> First note that we considered the imbalanced case in Section 3.1. More precisely, we considered what it would imply to first rescale the pictures so that they have unit mass. For large pictures and small noise, Theorem 1 makes a step towards understanding what this implies. In light of the reviewer’s suggestion, we have added an analysis of unbalanced OT in Appendix B.
>
> * _I am no expert in cryo-EM, but I would be interested in a discussion and maybe an empirical comparison with ‘learned perceptual metrics’_
>
> Computational pipelines in cryo-EM typically need to know when two macromolecules are close in terms of their composition and/or continuous degrees of freedom (e.g. open/close states, ratcheting motion, twisting, etc.). In light of this, perceptual metrics such as LPIPS are not natural choices and we are not aware of any papers that attempt to do this. Currently, the most common choice of metric for cryo-EM images is Euclidean, however optimal transport metrics and approximations are increasingly being used due to their natural connection to physical motion, e.g., physical transformations such as translation and rotation upper-bound the Wasserstein metric between a molecule and its transformed counterpart. We chose to focus on the latter, partly due to their increasing use in the cryo-EM community.

---

> > ### Comment · Reviewer_oX7e · 2025-11-26
> > **Thank you**
> >
> > Thanks. I keep my positive score :)

---

### Official Review · Reviewer_j41i · 2025-10-31

**Soundness:** 2
**Presentation:** 2
**Contribution:** 2
**Rating:** 0
**Confidence:** 5

**Summary:**

This paper provides a comprehensive theoretical and empirical investigation into the noise sensitivity of the Wasserstein metric for images under pixel-wise additive Gaussian noise. The authors derive non-asymptotic upper bounds on the error of the signed $p$-Wasserstein distance between clean and noisy images, showing in particular that the error scales sublinearly (as $\sqrt{\sigma}$) for $W_2$, in contrast to the linear scaling observed with the Euclidean ($L^2$) norm. The work includes rigorous proofs, synthetic experiments, experiments on cryo-EM datasets, and a detailed analysis of the phenomenon in which increasing noise can sometimes decrease the Wasserstein distance, supported by visualizations and quantitative benchmarks.

**Strengths:**

- The paper rigorously derives upper bounds for the influence of additive Gaussian noise on signed Wasserstein distances, with clear and carefully stated assumptions. The approach is anchored in state-of-the-art multiscale coupling arguments and careful handling of signed measures, extending prior work.
- The experiments validate theoretical scaling laws on standard image datasets and bridge the gap between theory and practice. The authors include detailed ablation studies on noise scaling, metric behavior across image types, and practical benchmarks in high-noise cryo-EM alignment tasks.
- The analysis and illustration of the "decreasing distance under increasing noise" phenomenon (Section 4.4, Figures 7–8) are insightful and highlight subtleties in OT metrics under sparsity and mass-bridging scenarios.

**Weaknesses:**

- While the paper derives upper bounds for noise sensitivity (notably for $W_2$ and other $W_p$ metrics), the practical tightness of these bounds is not systematically quantified, aside from qualitative observations (e.g., Figure 3 notes the lack of tightness for $W_2$ and $W_3$). A more explicit and systematic investigation—such as plots illustrating the gap between theoretical upper bounds and empirical measurements across parameter regimes, or commentary explaining where and why the bounds become loose—would substantially clarify the practical utility of the theory.

- Although some motivation is drawn from deep learning practice (e.g., references to WGAN and WAE), the paper lacks experimental or conceptual analysis of how the theoretical findings could inform model architectures, loss function design, or optimization routines in modern machine learning. For example, discussing implications for robustness in learned deep representations would strengthen the connection between theory and practice.

- Although the reviewer appreciates the theoretical justification for analyzing effect of noisy image with the signed Wasserstein distance, the paper’s format is incorrect (it lacks the statement “Under review as a conference paper at ICLR 2026” in the header). As a result, the reviewer must assign a score of 0 for formatting compliance.

**Questions:**

- Can the authors provide a more systematic analysis of the practical tightness of the derived upper bounds for noise sensitivity?

- What are the concrete implications of losing the metric properties (e.g., the triangle inequality) for $W_p^{\pm}$ with $p>1$ in empirical applications? Are there known failure cases or recommended best practices for using the signed $W_2$ in image similarity or learning scenarios?

- Given the growing use of OT-based losses in deep learning, could the authors elaborate on the implications of their findings for architecture or algorithm design—for example, in selecting loss functions for noisy training environments?

- Are there practical guidelines for choosing between $W_1$, $W_2$, or $L^2$ distances in specific image analysis contexts, based on the empirical or theoretical results presented in this work?

---

### Official Review · Reviewer_PgAu · 2025-11-01

**Soundness:** 2
**Presentation:** 2
**Contribution:** 2
**Rating:** 4
**Confidence:** 3

**Summary:**

The paper analyzes how the signed Wasserstein distance $W_p^\pm$ behaves when pixel-level noise is added to images. The authors derive non-asymptotic upper bounds on the deviation between noisy and clean distances. Experiments on synthetic data and cryo-EM projections validate the predicted scaling, and show that $W_2^\pm$ can preserve geometric structure better than $L_2$ in heavy noise. However, because the analysis remains confined to grid-based image histograms and does not engage with robust OT variants already in the literature, I do not see enough new insight to recommend acceptance.

**Strengths:**

- The authors leverage known dyadic decomposition tools (Weed \& Bach 2019) and adapt them to the signed OT setting, resulting in explicit, dimension-aware bounds that match the empirical scaling trends.
- Experiments are well aligned with the theory: the scaling plots, robustness study across different image families, and cryo-EM case study convincingly illustrate the practical message that $W_2^\pm$ can tolerate higher noise.

**Weaknesses:**

- All theory assumes images as discrete measures on a regular grid; there is no discussion of other data modalities or continuous supports. Many OT applications (e.g., WGANs) treat an image as part of a higher-dimensional distribution rather than as a histogram, so it is unclear how the analysis transfers.
- No lower bounds or tightness discussion. The results provide upper bounds on $\mathbb{E}\,W_p^\pm$ but do not show matching lower bounds or establish tightness beyond qualitative plots.
- Signed OT complications. For $p>1$ the signed Wasserstein cost is not a metric and lacks triangle inequality. The text acknowledges this but theorems such as Theorem~5 still leverage triangle-like arguments; more explanation is needed to ensure the statements are well-defined.
- Instability of Wasserstein distances under noise has already motivated partial and unbalanced OT formulations [1--4]; the paper neither positions its analysis relative to this literature nor compares $W_p^\pm$ against robust alternatives in experiments. Without such context it is hard to see the novelty or practical benefits.

References:
[1] Raghvendra, S. et al.. “A New Robust Partial $p$-Wasserstein-Based Metric for Comparing Distributions,” ICML 2024.
[2] Benamou, J.-D., et al “Iterative Bregman Projections for Regularized Transportation Problems,” SIAM Journal on Scientific Computing 2015.
[3] Chizat, S. et al. “Scaling Algorithms for Unbalanced Optimal Transport Problems,” Mathematics of Computation 2018.
[4] Chapel et al. "Partial Optimal Transport with Applications on Positive-Unlabeled Learning" NeurIPS 2020

**Questions:**

- Can you elaborate on why you inject additive zero-mean Gaussian noise that allows pixels to become negative? In practical pipelines one might clip or renormalize intensities—would your analysis or conclusions still hold under such nonnegative noise models, and what phenomena are lost if we do so?
- Theorem 5 set the optimal cost $W_p(\mu,\nu)$ with the bound terms. Could you provide a corollary specialized to the case $\mu=\nu$?
- For the alignment study, do you estimate the distances on raw values or after histogram equalization / normalization?
- Is it possible to extend the analysis to entropic OT distances (regularized Sinkhorn costs)?
- Can you provide empirical comparisons between $W_p^\pm$ and robust alternatives such as the robust partial $p$-Wasserstein metric [1] or unbalanced OT variants [3]?

---

> ### Author Response · Authors · 2025-11-24
> **Answer to the weaknesses section of the review**
>
> We thank the reviewer for their careful reading of our paper and thoughtful commentary. We have made several improvements and fixes to the paper in light of the review. A point-by-point response follows:
>
> * _All theory assumes images as discrete measures on a regular grid; there is no discussion of other data modalities or continuous supports. Many OT applications (e.g., WGANs) treat an image as part of a higher-dimensional distribution rather than as a histogram, so it is unclear how the analysis transfers._
>
> This is not the setting we considered. What you suggest to look at is already well understood and requires different tools. In the case that you seem to have in mind, two phenomena come into play: first, the approximation of the distribution of $I_i+\varepsilon_i$ by a discrete measure and second, the approximation of a measure with its convolved counterpart.
> The rate in Wasserstein $p$ distance follows from existing results on the sample complexity of optimal transport and on convolution of measures. There is a plethora of works addressing the first problem, see the reference below for one such work.
> For the second problem of computing the optimal transport distance between a measure and its convolved counterpart the following argument gives an upper bound.
> Consider  $X\sim \mu,Y\sim \nu, Z\sim \gamma$ three independent random variables, $\gamma$ being the convolution measure. As  $\mathbb{E} [ \|X -Y+ Z\|^2] \le \mathbb{E} [ \|X -Y\|^2] + \mathbb{E} [ \|Z\|^2]$, one can easily construct an upper bound on the optimal transport cost. Below is a reference that studies this kind of question.
>
> Hundrieser, Shayan, Thomas Staudt, and Axel Munk. "Empirical optimal transport between different measures adapts to lower complexity." In Annales de l'Institut Henri Poincare (B) Probabilites et statistiques, vol. 60, no. 2, pp. 824-846. Institut Henri Poincaré, 2024.
>
> Ding, Yunzi, and Jonathan Niles-Weed. "Asymptotics of smoothed Wasserstein distances in the small noise regime." Advances in Neural Information Processing Systems 35 (2022): 19203-19214.
>
> * _No lower bounds or tightness discussion. The results provide upper bounds on $\mathbb{E}W_p^\pm$ but do not show matching lower bounds or establish tightness beyond qualitative plots._
>
> This is an excellent idea. We have added a lower bound to $W_1$ in Theorem 2. We have also tried to develop the same type of bound for $W_p$. As the latter are not sharp, we did not include them in the revised manuscript.
>
> * _Signed OT complications. For $p>1$ the signed Wasserstein cost is not a metric and lacks triangle inequality. The text acknowledges this but theorems such as Theorem~5 still leverage triangle-like arguments; more explanation is needed to ensure the statements are well-defined._
>
> We understand the concern of the reviewer. Our proof indeed does not assume a triangle inequality. As detailed in the appendix (specifically Equations 103-108), our derivation is strictly built on the concavity of $t^{1/p}$, the linearity of expectation, the Jensen inequality and the result in Theorem 2.  We have carefully written the proof keeping in mind that the triangle inequality does not hold and thus we did not use it directly, nor did we use any theorem that relies on it.
>
> * _Instability of Wasserstein distances under noise has already motivated partial and unbalanced OT formulations [1--4]; the paper neither positions its analysis relative to this literature nor compares $W_p^\pm$ against robust alternatives in experiments. Without such context it is hard to see the novelty or practical benefits._
>
> We agree that unbalanced OT is an important related notion and in light of this comment have added a sentence to the Related Work section. However, to the best of our knowledge, the combination of unbalanced transport with signed measures had not been considered in the literature thus far. Our aim in this paper is to understand the behavior of more mainstream approaches.
> In the revised manuscript, we considered applying an unbalanced version of optimal transport to the Jordan decomposition and rearrangement of measures in Appendix B, where we establish a bound in that case.

---

> ### Author Response · Authors · 2025-11-24
> **Answers to the questions raised by the reviewer**
>
> * _Can you elaborate on why you inject additive zero-mean Gaussian noise that allows pixels to become negative? In practical pipelines one might clip or renormalize intensities—would your analysis or conclusions still hold under such nonnegative noise models, and what phenomena are lost if we do so?_
>
> We wanted to consider a simple setting that is as close as possible to the classic Gaussian i.i.d. model without adding technical complications such as clipping. We are indeed motivated by high-noise applications in the field of cryo-EM, where the noise looks Gaussian, negative values are common, and typically no clipping is performed. The zero-mean Gaussian noise model that we consider is equivalent to drawing i.i.d. Gaussian values and then subtracting the mean over the pixels of the latter noise values. We have added a remark stressing this in the paper, right after Assumption 1. Note also that renormalization is considered in Section 3.1.
> For non-negative noise models, as in Poisson noise, we believe one could do a similar analysis. The signals would have to be normalized, or unbalanced OT would need to be incorporated in that case. Positivity of the noise values suggests that the signed Wasserstein distance would not be required in most practical cases.
>
> * _Theorem 5 set the optimal cost $W_p(\mu, \nu)$  with the bound terms. Could you provide a corollary specialized to the case $\mu=\nu$?_
>
> We thank the reviewer for this suggestion. We have extended Theorems 2 and 3 to include two different sources of noise to cover the case of two sources of noise.
>
> * _For the alignment study, do you estimate the distances on raw values or after histogram equalization / normalization?_
>
> Yes, we normalize images to sum up to 1. We added an explicit mention of this in the paper in subsection 4.3, just below Figure 4.
>
> * _Is it possible to extend the analysis to entropic OT distances (regularized Sinkhorn costs)?_
>
> This is an interesting idea. However, the proof technique based on multiscale bounds does not seem to combine well with the entropy functional. We leave it for future research.
>
> * _Can you provide empirical comparisons between $W_p^\pm$  and robust alternatives such as the robust partial p-Wasserstein metric [1] or unbalanced OT variants [3]?_
>
> These variants of OT have been proposed to be robust against mass imbalance. It is thus a different notion of robustness than the one we consider in the paper. Nonetheless we have added a paragraph about this under Related Work in the revised manuscript.
>
> We believe we have adequately addressed the concerns of the referee and have made several improvements to the manuscript following the suggestions in the review. In light of this, we believe the ratings of the paper should be increased.

---

### Meta-Review · Area_Chair_DPiw · 2025-12-25

**Summary:**

The authors consider the noise sensitivity of the signed Wasserstein distance problem for pixel-wise additive noise in images, and derive non-asymptotic upper bounds for the deviation.

The Reviewers think that the theoretical finding results could be interested in for the community. However, the Reviewers also raised some critical concerns on the limitation of problem setup (e.g., additive zero-mean Gaussian noise assumptions); theoretical analysis on the tightness.

Overall, we think the submission falls short to the bar. Some critical concerns from the Reviewers are relevant. The authors may take into account some suggestions from the Reviewers to improve the submission.

**Reviewer Concerns:**

The Reviewers have some following concerns:

+ Reviewer PgAu: limit to grid-based image histograms; lack theoretical lower bounds, analysis on tightness; metric properties for signed Wasserstein with $p > 1$; relation to robust OT variants under various noisy settings; additive zero-mean Gaussian noise with negative-value image pixels for problem setting.

+ Reviewer j41i: theoretical analysis on tightness; practical impacts of the theoretical findings; paper format; empirical effects of lacking metric properties for signed Wasserstein with $p > 1$.

+ Reviewer oX7e: related works; niche problem; presentation; additive zero-mean Gaussian noise assumption

**Reviewer Scores:**

The authors have added a lower bound in Theorem 2; clarified the proof for signed Wasserstein with $p > 1$ on its lack of triangle inequality; cryo-EM application domain; different settings for robustness; paper format; related works for the revision.

The authors partially address concerns from the Reviewers about the presentation. It is also a plus to provide an additional lower bound in Theorem 2. However, the assumption on additive zero-mean Gaussian noise is a clear limitation, we think it is better to relax it with the unbalanced optimal transport approach. It is also important to theoretically analyze the tightness for theoretical finding results.

---

### Decision · Program_Chairs · 2026-01-26

Reject